# Diverse synaptic and dendritic mechanisms of complex spike burst generation in hippocampal CA3 pyramidal cells

Snezana Raus Balind[1], Ádám Magó[1,2], Mahboobeh Ahmadi[1], Noémi Kis[1], Zsófia Varga-Németh[1], Andrea Lőrincz[3] & Judit K. Makara[1]

Complex spike bursts (CSBs) represent a characteristic firing pattern of hippocampal pyramidal cells (PCs). In CA1PCs, CSBs are driven by regenerative dendritic plateau potentials, produced by correlated entorhinal cortical and CA3 inputs that simultaneously depolarize distal and proximal dendritic domains. However, in CA3PCs neither the generation mechanisms nor the computational role of CSBs are well elucidated. We show that CSBs are induced by dendritic $Ca^{2+}$ spikes in CA3PCs. Surprisingly, the ability of CA3PCs to produce CSBs is heterogeneous, with non-uniform synaptic input-output transformation rules triggering CSBs. The heterogeneity is partly related to the topographic position of CA3PCs; we identify two ion channel types, HCN and Kv2 channels, whose proximodistal activity gradients contribute to subregion-specific modulation of CSB propensity. Our results suggest that heterogeneous dendritic integrative properties, along with previously reported synaptic connectivity gradients, define functional subpopulations of CA3PCs that may support CA3 network computations underlying associative memory processes.

[1] Laboratory of Neuronal Signaling, Institute of Experimental Medicine, Hungarian Academy of Sciences, 43 Szigony Str., 1083 Budapest, Hungary. [2] János Szentágothai School of Neurosciences, Semmelweis University, 1085 Budapest, Hungary. [3] Laboratory of Cellular Neurophysiology, Institute of Experimental Medicine, Hungarian Academy of Sciences, 43 Szigony Str., 1083 Budapest, Hungary. Correspondence and requests for materials should be addressed to J.K.M. (email: makara.judit@koki.mta.hu)

The hippocampus is thought to play a fundamental role in spatial and contextual memory, with each hippocampal area (dentate gyrus (DG), CA3, CA2, and CA1) contributing at different steps of information processing. Elucidating the cellular and circuit mechanisms underlying these locally specialized network operations is essential for understanding hippocampal computations as a whole.

A longstanding theory about neuronal coding is that, beside firing rate, the temporal characteristics of firing also carries information. Specifically, high-frequency trains of action potentials (APs), i.e., bursts are thought to represent an information-rich communication signal enabling reliable neurotransmission and increasing coding capacity[1,2]. The phenotype and mechanism of bursting are diverse among neuron types[2,3]. A specific burst form is the complex spike burst (CSB), a hallmark firing behavior of hippocampal PCs during navigation and sharp-wave ripples (SWRs)[4–8], which consists of 2–6 APs with progressively declining amplitude and increasing duration. CSBs were recently connected to place field formation and remapping by CA1PCs. Spontanous or artificial induction of CSBs converted silent CA1PCs to place-coding cells[6,9,10], and stimulation triggering CSBs induced formation of new place fields in active PCs[10], with new fields appearing at the location where the CSBs occurred. Burst propensity of CA1PCs has also been linked to subsequent place coding in a novel environment[11]. Thus, CSBs may represent a specific form of cellular information processing that can induce rapid changes in feature coding of individual hippocampal neurons.

The synaptic and intrinsic mechanisms of bursting can be variable, and thereby the functional code a burst represents may depend on the cell type, system and context[2]. Generation of CSBs was extensively studied in hippocampal CA1 and cortical layer 5 PCs in vitro[12,13]. In these cell types (typically having single or twin apical dendritic trunk) CSBs result from an interplay of synaptic inputs and voltage-dependent conductances (VDCs) in two spatially separate (somatic and apical dendritic) spike initiation zones. Dendritic depolarization by correlated synaptic input and/or backpropagating APs (bAPs) opens voltage-gated $Ca^{2+}$ channels (VGCCs) producing further regenerative depolarization, which propagates to the soma and evokes additional APs. Regenerative recruitment of NMDARs further contributes to produce a prolonged dendritic $Ca^{2+}$/NMDA spike, also called plateau potential. In CA1PCs in vitro, dendritic plateaus are triggered by co-activation of proximal (Schaffer collaterals from CA3) and distal apical (from entorhinal cortex (EC) layer 3) excitatory synapses, which alone do not efficiently induce plateaus[13]. At the soma, the plateau appears as an afterdepolarization (ADP) that lasts ~20–60 ms and drives the burst[13]. Based on these results, CSBs were proposed to represent associative dual-pathway input patterns (combining current and stored contextual information) transformed into a specific bursting AP output, that may initiate stable location coding presumably by inducing long-lasting plasticity of synaptic strength[9] and/or cellular excitability.

Despite their important role in the hippocampal microcircuit, much less is known about the synaptic and dendritic mechanisms of CSB generation in CA3PCs. CA3PCs often fire CSBs in vivo[7,8,14], under some circumstances even more so than CA1PCs[14,15]. In vitro experiments[16–19] and simulations[20,21] suggested that dendritic VDCs (including VGCCs) are involved in CSB generation in these cells, but a detailed investigation remained lacking.

The morphological, synaptic and electrophysiological properties of CA3PCs differ from CA1PCs in several aspects. CA3PCs receive three main, spatially segregated excitatory input types: inputs from the DG via mossy fibers (MF) target thick proximal dendrites forming synapses with complex spines (thorny excrescences);

recurrent collateral (RC) synapses from other CA3PCs arrive to thinner higher-order basal and proximal apical dendrites; and inputs from EC layer 2 target distal apical branches. In contrast to CA1PCs, the main apical dendrite of CA3PCs typically branches in stratum (str.) lucidum into multiple trunks. Although the electrical properties of CA3PC dendrites are less well characterized, recent studies revealed that they can produce local $Na^+$ and NMDAR-mediated dendritic spikes (d-spikes)[22–24]. Early recordings also indicated occurrence of VGGC-mediated dendritic $Ca^{2+}$ spikes[18]; however, neither the properties of $Ca^{2+}$ spikes nor the generation of plateau potentials in CA3PC dendrites have been explored. Importantly, since topographic variability of morphological, genetic, synaptic and electrophysiological properties of PCs in CA3 has long been recognized[25–33], the cellular mechanisms contributing to CSB generation may also be heterogeneous.

Here we set out to elucidate the synaptic and dendritic mechanisms of CSB generation in CA3PCs in the adult rat hippocampus using in vitro patch-clamp recordings and two-photon imaging and uncaging techniques. We first show that CSBs are generated by dendritic, mainly VGCC-mediated spikes upon somatic or synaptic depolarization. Second, the ability of CA3PCs to produce CSBs is highly variable, which is partly related to their topographic position. Third, we identify two ion channel types, HCN and Kv2 channels, whose proximodistal activity gradients contribute to subregion-specific modulation of CSB propensity.

## Results

**Diversity of CSB propensity in CA3PCs.** To investigate the contribution of distal apical dendrites of CA3PCs to complex spike bursting, we loaded neurons with a $Ca^{2+}$ sensitive fluorescent dye (either OGB-1 or OGB-6, 100 μM) and a $Ca^{2+}$-insensitive fluorescent marker (Alexa Fluor 594, 50 μM) via a somatic patch pipette. We first set out to determine the CSB propensity of CA3PC in response to a series of moderate somatic depolarizing current injections ($I_{inj}$, five 100-ms-long pulses of 300–600 pA with 80.55-ms-long interpulse intervals) from ~−70 mV baseline $V_m$. In many CA3PCs even relatively small $I_{inj}$ triggered not only simple APs, but also CSBs that manifested as a characteristic burst of ≥2 APs with progressively smaller amplitude and longer duration, riding on a prolonged ADP of variable duration (~20–70 ms, often terminated by or outlasting the end of the step; Fig. 1a, b; Methods). CSBs (but not simple APs) were typically accompanied by large time-locked $Ca^{2+}$ response even in distal apical dendrites in str. lacunosum-moleculare (SLM) in an all-or-none fashion (Fig. 1a–d).

We quantified CSB propensity by calculating the CSB rate, defined as the number of pulses with CSB divided by the total number of pulses (see Methods section). The CSB rate increased with stronger $I_{inj}$ (Fig. 1e, f, $p < 0.001$, Wilcoxon test). Within the pulse series, CSBs were most likely triggered on the first pulse, but often appeared also on later pulses, especially upon stronger $I_{inj}$ (Fig. 1e). In general, CSB rate in CA3PCs was stronger than in CA1PCs (Fig. 1f, Supplementary Fig. 1 for CA1PCs; at 300 pA: $p = 0.008$, at 600 pA: $p < 0.001$, $n = 118$ CA3 and 26 CA1 PCs, Mann–Whitney test).

We observed a remarkable heterogeneity among CA3PCs in their CSB propensity: some cells generated CSBs already at low $I_{inj}$ (300 pA, Fig. 2a, see also Methods), whereas other cells did not show CSBs even upon strong $I_{inj}$ (600 pA, Fig. 2b), and a minority of neurons had intermediate CSB threshold (Fig. 2c, d). While the threshold of neurons to fire CSBs was distributed in a wide range, the CSB rate distribution was non-gaussian at both 300 and 600 pA current injections (Shapiro-Wilks test, $p < 0.001$ for both) and formed distinct populations (Fig. 2c). Therefore, we operationally sorted the recorded neurons into the following

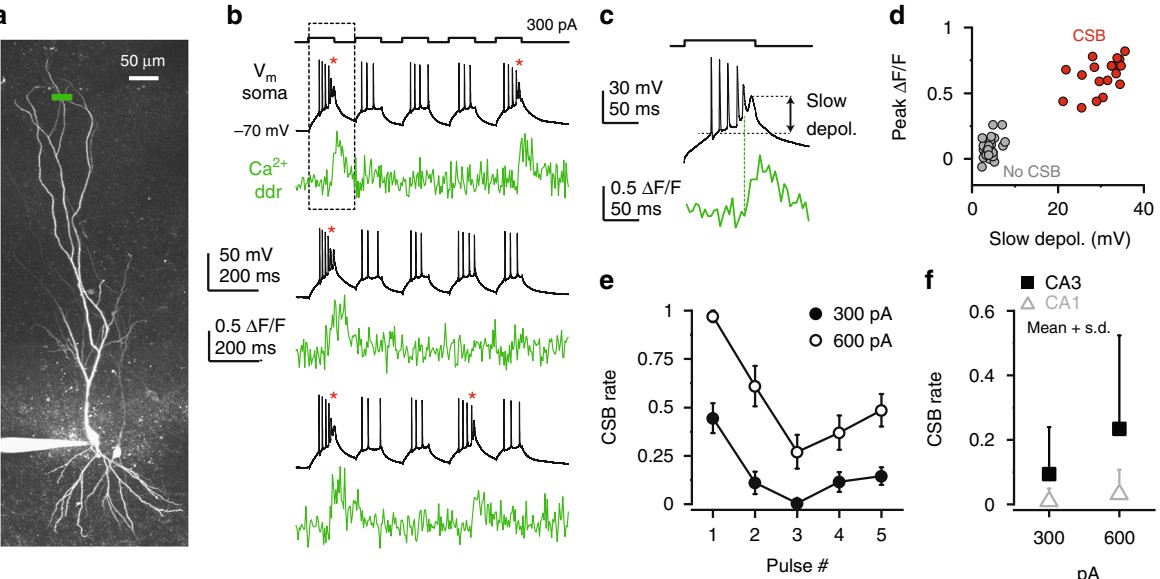

**Fig. 1** Complex spike bursts in CA3PCs. **a** Z-stack of a CA3PCs loaded with 50 μM Alexa594 and 100 μM OGB-6F. $Ca^{2+}$ imaging line is indicated in green. **b** Three representative somatic current-clamp and corresponding dendritic (ddr) $Ca^{2+}$ traces recorded in the cell shown in **a**. Complex spike bursts are marked by red asterisks. Current injection protocol is shown on top. **c** Magnified traces from the dashed box in **b**, illustrating the slow depolarization underlying CSB and the time-locked dendritic $Ca^{2+}$ signal. **d** Results of the total 50 pulses of 10 traces recorded from the cell shown in **a–c**. Note the separation of pulses with and without CSBs. **e** CSB rate on different pulses in the train. A set of 20 cells that produced CSBs at 300 pA $I_{inj}$ were included in the analysis. Symbols and error bars represent mean ± s.e.m. **f** CSB rate at 300 and 600 pA $I_{inj}$ in CA3PCs ($n = 118$ and 117 cells, respectively) and CA1PCs ($n = 26$ cells). Symbols and error bars in **f** represent mean + s.d. Source data are provided as a Source Data file

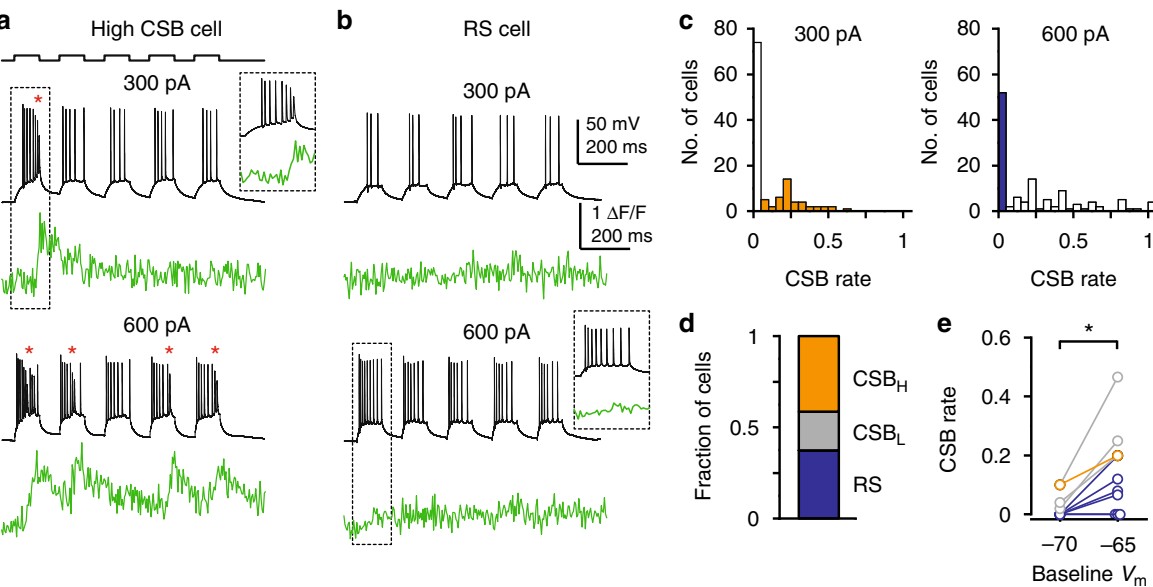

**Fig. 2** Diversity of CSB propensity among CA3PCs. **a** Representative voltage and distal dendritic $Ca^{2+}$ responses of a high CSB cell to repetitive 100-ms somatic injection of 300 and 600 pA (protocol on top). Dashed box is shown enlarged in inset. CSBs are indicated by red asterisks. **b** Same as in **a** for a representative RS cell. Note the lack of CSBs at 600 pA $I_{inj}$. **c** Histogram of CSB rate by 300 pA (left) and 600 pA (right) $I_{inj}$. $CSB_H$ cells and RS cells are indicated in orange and blue, respectively. **d** Fraction of $CSB_H$, $CSB_L$, and RS cells in the whole recorded dataset. **e** Effect of $V_m$ on CSB rate by 600 pA $I_{inj}$. Color code as in **d**. *$p < 0.05$. Source data are provided as a Source Data file

categories: cells with high CSB propensity ($CSB_H$; $I_{inj}$ threshold ≤ 300 pA), cells with low CSB propensity ($CSB_L$; $I_{inj}$ threshold 400–600 pA) and regular spiking cells (RS; $I_{inj}$ threshold > 600 pA; Fig. 2d). Cells with different CSB rates were found in slices from the same animals (Supplementary Fig. 2a). Depolarization increased CSB rate in most cells (Fig. 2e, Wilcoxon test, $p = 0.011$, $n = 11$), including some, but not all RS cells. Despite

the large heterogeneity in CSB propensity, the properties of the first AP evoked by 600 pA $I_{inj}$ were similar across the three CA3PC groups (Supplementary Fig. 2), with a slightly more negative AP threshold in $CSB_H$ cells (Supplementary Fig. 2g). Although the range of first AP onset time largely overlapped across the three CA3PC groups, it was significantly shorter in $CSB_H$ cells (Supplementary Fig. 2h).

**Intrinsic generation of CSBs via dendritic Ca$^{2+}$ spikes.** In recurrent circuits such as the CA3 area, network activity may generate bursts[3]. However, blockade of neither AMPA/NMDA glutamate receptors (Fig. 3a, $n = 10$, $p = 0.753$, Wilcoxon test) nor GABA$_A$/GABA$_B$ receptors (Fig. 3b, $n = 11$, $p = 0.892$, Wilcoxon test) influenced CSB rate, indicating that under our conditions CSBs were generated by an intrinsic cellular mechanism. Indeed, in the presence of the voltage-gated Na$^+$ channel blocker TTX, the $I_{inj}$ protocol (although sometimes with stronger $I_{inj}$ up to 1200 pA) evoked regenerative depolarizations that resembled spikes mediated by VGCCs in microelectrode experiments[17,18] (Fig. 3c, $n = 11$ cells). These regenerative responses were eliminated by 100–200 μM Cd$^{2+}$ (Fig. 3c, d, $n = 7$, $p = 0.017$, Wilcoxon test), also implying a role for VGCCs in their generation.

To explore the generation site of Ca$^{2+}$ spikes, we measured Ca$^{2+}$ in different subcellular compartments while applying 1-second-long $I_{inj}$ in TTX (Fig. 3e, f). The long depolarizing step triggered full-blown regenerative events at a certain $I_{inj}$ threshold in almost all CA3PCs. Ca$^{2+}$ spikes showed variable amplitudes and kinetics across different CA3PCs, but had stereotypic phenotype within a given neuron (Fig. 3f). Ca$^{2+}$ spikes were accompanied by large, time-locked, steeply rising Ca$^{2+}$ signals in distal dendrites. In contrast, smaller spike-associated Ca$^{2+}$ signals—superposed on a ramping Ca$^{2+}$ rise evoked by the depolarization step—were observed at or near the soma, and no Ca$^{2+}$ signals were detected in axonal boutons (Fig. 3f, g, $n = 6$, Friedman ANOVA $p = 0.002$).

To further investigate the potential dendritic origin of the slow depolarization driving CSBs, we patched dendritic trunks of CA3PCs in proximal str. radiatum (~100–200 μm from the soma, $163 \pm 11$ μm, mean ± s.e.m.). In 8 of 10 recordings, $I_{inj}$ pulses applied through the dendritic patch pipette evoked not only fast bAPs (consistent with previous data[22], Fig. 3i) but also CSB-like voltage responses, characterized by large sustained depolarizations following individual bAPs (ADP$_{AP}$). The maximal ADP$_{AP}$ amplitude during the dendritically evoked and recorded CSBs ($11.9 \pm 1.6$ mV, $n = 8$, Fig. 3h, i) was larger than that measured during somatically evoked and recorded CSBs in interleaved experiments from other CA3PCs ($5.6 \pm 0.4$ mV, $n = 13$, $p < 0.001$, Mann–Whitney test, Fig. 3h, i), consistent with a dendritic origin of the slow ADP. In contrast, AP amplitude was smaller in dendrites (Fig. 3i, $p < 0.001$, Mann–Whitney test), as expected from a signal propagating from the soma[22]. After bath application of TTX, dendritic $I_{inj}$ (450–900 pA) evoked isolated Ca$^{2+}$ spikes (Fig. 3h, tested in $n = 5$ cells).

Taken together, our results indicate that relatively long-lived somatic or proximal dendritic depolarization can trigger VGCC-mediated d-spikes in CA3PCs, which likely underly CSB firing.

**Synaptic input patterns evoking CSBs.** In CA1PCs it was proposed that correlated activation of distal (EC) and proximal (SC) synapses is required for VGCC-mediated and NMDA-mediated dendritic plateau potentials driving CSBs, with properties similar to the CSBs we recorded in CA3PCs[6,13]. Given the unusually high CSB propensity of many CA3PCs, the question arises: what types of spatiotemporal input patterns can evoke CSBs in CA3PCs, and are the rules of input–output transformation similar in CA3PCs with different CSB propensities? To address these questions we first exploited the precise spatiotemporal control of postsynaptic activation by two-photon glutamate uncaging[23] to investigate the effect of correlated activity of colocalized RC or EC synaptic inputs. We selected 15–20 spines on short segments of thin dendrites in the basal, apical oblique or tuft arborizations, and stimulated them quasi-synchronously 5 times at 40–50 Hz (gamma burst, see Methods section). Stimulus strength was set to produce peak compound gluEPSP amplitudes near AP threshold. This local stimulation failed to trigger CSBs in RS cells, but often induced CSBs in CSB$_H$ cells (Fig. 4a–c), with intermediate phenotype in CSB$_L$ cells (Fig. 4c). We found positive correlation between CSB propensities by somatic $I_{inj}$ and by dendritic stimulation with uncaging (Fig. 4c, Spearman $R = 0.849$, $p < 0.001$, $n = 46$ dendrites from 31 cells). The synaptic stimulation applied in basal or apical oblique dendrites often also evoked regenerative Na$^+$ and NMDAR-mediated spikes, similarly to previous reports[22–24] (Supplementary Fig. 3). CSB propensity by synaptic stimulation was promoted by NMDAR activation: bath application of D-AP5 (100 μM) moderately decreased the occurrence rate of (but did not eliminate) CSBs evoked by uncaging (Fig. 4d, $n = 8$ cells, $p = 0.043$, Wilcoxon test), in addition to an expected reduction of EPSP summation (EPSP$_{peak}$/EPSP$_{1st}$: from $5.28 \pm 0.53$ to $3.58 \pm 0.33$, $n = 8$, $p = 0.011$, Wilcoxon test). Thus, depending on the excitability of dendrites to produce VGCC-mediated spikes, even local, unimodal synaptic activity can generate CSBs in many CA3PCs.

Next we asked whether RS cells—which failed to produce CSBs by local input or by somatic depolarization alone (Fig. 4b, c)—can generate CSBs upon combined proximal-distal depolarization. To this end, we stimulated distal synapses via a theta pipette placed in SLM in a theta-gamma burst fashion (5× at 50 Hz, trains repeated 5× at 5.5 Hz) in the presence of GABA$_A$ and GABA$_B$ receptor blockers. The stimulus produced ~10 mV peak somatic EPSPs (range: 4–15 mV), but no Ca$^{2+}$ signals were evoked in the dendrites situated closest to the theta pipette, indicating spatially broad rather than localized synaptic activation. Synaptic stimulation was performed either alone, or combined during each stimulus with somatic $I_{inj}$ (600 pA $I_{inj}$) as a proxy for simultaneous perisomatic depolarization[6] (e.g., by burst activation of proximal RC and/or MF inputs) (Fig. 4f). Distal synaptic stimulation did not efficiently evoke CSB in RS cells; however, when synaptic stimulation was combined with proximal depolarization, CSBs were evoked in the majority of RS cells (Fig. 4f, g, $n = 8$, $p = 0.003$ for CSB rate, Friedman test; combined stimulus differs from $I_{inj}$ or ES alone, $p < 0.05$ Wilcoxon test). Thus, RS cells are capable of producing CSBs, but under our experimental conditions do so only during sufficiently strong combined proximodistal depolarization, which naturally requires correlated activation of different (spatially segregated) input pathways. Altogether, the results indicate that the rules to generate dendritic VGCC/NMDAR-mediated spikes are not uniform among CA3PCs, suggesting diverse cellular and computational roles of these d-spikes in CA3.

**Topographic and morphological correlates of CSB propensity.** What are the mechanisms behind the heterogeneous capacity of CA3PCs to generate CSBs? Since several properties of CA3PCs vary according to their topographic location[26–37], we first sought to elucidate the relation of CSB propensity with soma position. We sorted those dorsal CA3PCs where we had a record of both proximodistal (Fig. 5a) and radial (Fig. 5b) soma position into six groups. For proximodistal position we defined three regions (a-b-c; equal one-thirds of the full proximodistal length[33]). For radial position we categorized cells as superficial (soma within str. pyramidale) or deep (soma in str. oriens or on the pyramidale-oriens border).

We found a complex pattern of CSB propensity as a function of both parameters (Fig. 5c–f). We analyzed the results with two-way ANOVA, using data with 600 pA $I_{inj}$ in deep and superficial CA3a and CA3c cells ($p = 0.027$ for proximodistal axis, $p = 0.173$ for radial axis, $p = 0.025$ for interaction). CSB rate and the ratio of CSB cells increased along the proximodistal axis (c-to-a;

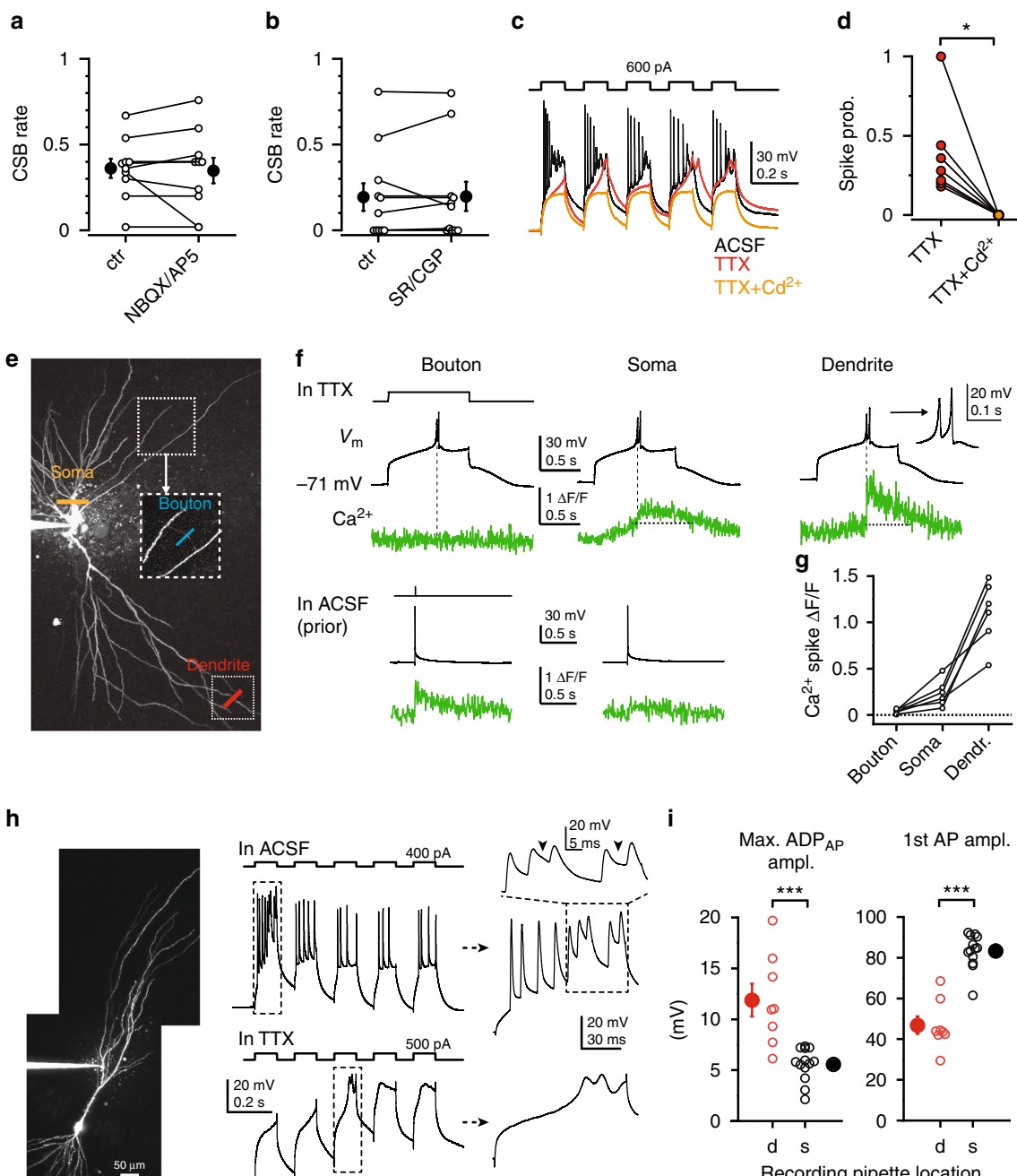

**Fig. 3** CSBs are produced intrinsically by VGCC-mediated d-spikes. **a** Effect of AMPA/NMDA receptor inhibitors (10 μM NBQX and 50 μM D-AP5; tested on CSB cells) on CSB rate at the same $I_{inj}$ strength (300 or 600 pA). **b** Effect of GABA$_A$/GABA$_B$ receptor inhibitors (2 μM SR95531 and 2 μM CGP55845) on CSB rate at the same $I_{inj}$ strength (300 or 600 pA). **c** Representative voltage responses of a CSB$_H$ cell under control conditions (black), in the presence of 1 μM TTX (red), and after the subsequent addition of 200 μM $Cd^{2+}$ (orange). Note the regenerative responses in TTX that are completely eliminated by $Cd^{2+}$. **d** Summary of the effect of 100–200 μM $Cd^{2+}$ on spikes in TTX. **e** Z-stack of a RS cell loaded with 100 μM OGB-1 and 50 μM Alexa 594. Subsequent somatic, dendritic and bouton imaging locations are indicated by lines. **f** Top, example voltage and $Ca^{2+}$ recordings from the locations in **e**. Voltage protocol shown on top, left. Inset shows the temporal details of the spike. Bottom: $Ca^{2+}$ responses were evoked by single action potentials in the bouton and soma prior to TTX washin. **g** Comparison of the spike-related $Ca^{2+}$ signal amplitude (using OGB-1) in bouton, soma and dendrite of the same neurons. **h** Left, z-stack of a CA3PC patched on the apical dendrite. Middle, voltage traces recorded from the dendrite in ACSF (top) and after TTX application (bottom) applying repetitive $I_{inj}$ protocol. Right, pulses magnified from the dashed boxes in middle panels. Note the sustained depolarization (arrowheads) following some of the bAPs. **i** Maximal ADP$_{AP}$ amplitude (left) and amplitude of the first AP (right) measured in dendritic (d) and somatic (s) recordings (in different experiments). Filled symbols and error bars represent mean ± s.e.m. *$p < 0.05$, ***$p < 0.001$. Source data are provided as a Source Data file

$p = 0.027$, Fig. 5c–f). Differences between superficial and deep CA3PCs depended on proximodistal position: while CSB rate of superficial cells was independent of subregion, deep cells in CA3a tended to have higher CSB rate ($p = 0.065$, Tukey's post hoc test) and contained more CSB cells ($p = 0.034$, $\chi^2$ test) than the superficial population, whereas in CA3c there was no significant difference (Fig. 5c, CSB rate: $p = 0.909$, Tukey's test; ratio of CSB cells, $p = 0.424$, $\chi^2$ test). Basic electrophysiological

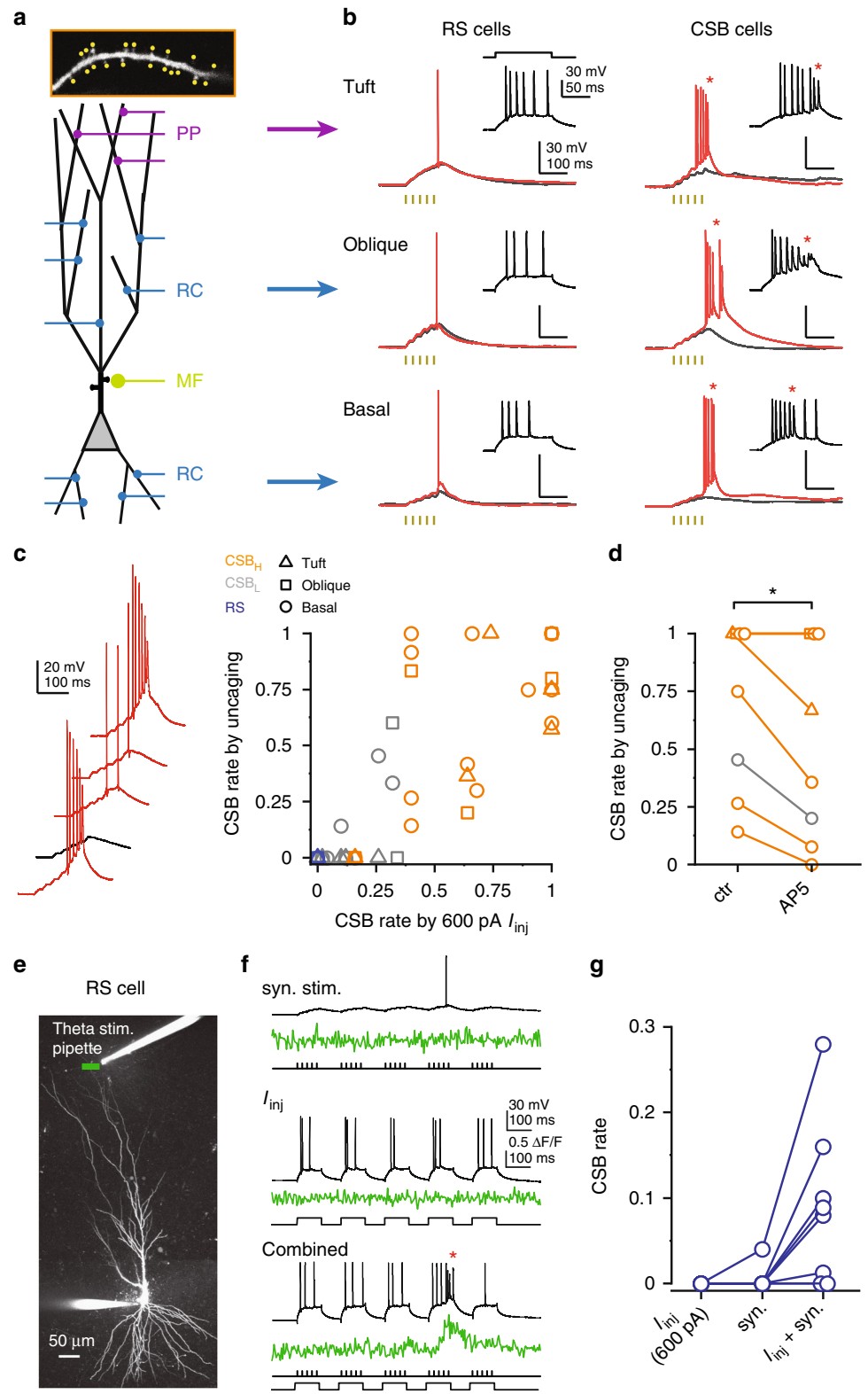

properties (resting $V_m$, $R_{in}$) did not seem to simply account for position-dependent differences in CSB propensity, although $R_{in}$ showed an increase in the a-to-c[33] and deep-to-superficial direction (Fig. 5d, e, two-way ANOVA, $p = 0.004$ for proximo-distal axis, $p = 0.005$ for radial axis, $p = 0.028$ for interaction). Notably, both RS and CSB cells could be found in all position groups.

We next examined whether dendritic morphology was related to CSB propensity. We compared several dendritic parameters of CA3PCs of the two extreme electrophysiological phenotypes (CSB$_H$ and RS cells), analyzed separately within CA3a and CA3c without distinction by radial position (Fig. 5g–j). The primary apical trunk was longer in CSB$_H$ cells than in RS cells, but only in CA3a and not in CA3c (two-way ANOVA, $p < 0.001$ for

**Fig. 4** Synaptic patterns evoking CSBs in different CA3PC types. **a–c** 2P glutamate uncaging experiments. **a** Top, dendritic segment illustrating the typical arrangement of stimulated synapses. Bottom, schematic of different input sources. **b** Example responses of RS cells (left) and CSB cells (right) to local uncaging at tuft (top), oblique (middle) and basal (bottom) dendrites. Black traces: subthreshold to AP, red traces: suprathreshold to APs. Insets show the cells' responses to 600 pA $I_{inj}$ (scale bar applies to all insets). Each example is from a different CA3PC. **c** Left, five responses evoked with the same uncaging stimulus in a $CSB_H$ cell. Subthreshold response is in black. Right, summary of the relationship of CSB rate to uncaging vs somatic depolarization in different CA3PC and dendrite types. $n = 46$ dendrites from 31 cells (RS = 10/6, $CSB_L = 13/9$, $CSB_H = 23/16$). Note that calculation of CSB rates differ for the two types of stimuli. **d** Uncaging evoked CSB rate before (ctr) and 10 min after bath application of D-AP5 (100 μM). Color and symbol codes as in **c**. **e–g** Axonal stimulation experiments. **e** Z-stack of a CA3PC, with theta pipette and imaging line position. **f** Somatic $V_m$ and dendrite $Ca^{2+}$ recordings from the cell in **e** by synaptic stimulation alone, somatic depolarization (600 pA) alone and the two combined in the presence of $GABA_A$ and $GABA_B$ receptor blockers (2 μM SR95531 and 2 μM CGP55845). **g** Summary of data recorded from RS cells. *$p < 0.05$. Source data are provided as a Source Data file

subregion, $p = 0.082$ for firing type, $p = 0.011$ for interaction, significant paired comparisons by post hoc Tukey test are indicated in Fig. 5i). This is consistent with dependence of CSB rate on soma position (Fig. 5c), because the primary apical trunk of deep CA3PCs runs longer to reach str. lucidum where it typically branches. Furthermore, we analyzed dendritic arbor complexity in a subset of fully reconstructed CA3PCs. Arbors were highly variable in all cell groups. Sholl analysis recapitulated some of the previously described differences between PCs in CA3a and c (Fig. 5h). However, we did not find drastic differences in total dendritic length, branchpoint numbers or distance of oblique dendrites from soma between the two electrophysiological phenotypes (Fig. 5j), although there was a trend for apical obliques to branch from the trunk more distally in RS cells. Altogether, while primary dendrite length was a predictor for bursting in distal CA3[34,37], in general dendritic morphology did not obviously predict CSB rate throughout the whole CA3 region.

A recent report[37] proposed that bursting CA3PCs located in deeper layers of CA3a-b comprise a novel CA3PC class that lacks thorny excrescences. Therefore, we retrospectively examined our $CSB_H$ cells in deep CA3a-b, of which high-magnification images of proximal apical dendrites were obtained. We confirmed that all of these $CSB_H$ cells ($n = 11$) had at least one (usually several) large, lobular, complex spine-like postsynaptic structure[38–40] on first- or higher order trunks in str. lucidum (Supplementary Fig. 4), inferring that they were not athorny CA3 cells. Although we do not exclude that athorny cells exist in the adult rat CA3, we conclude that a majority of thorny PCs in deep distal CA3 expresses complex spike bursting phenotype.

**Subregion-specific ion channels modulate CSB propensity.** We next investigated intrinsic electrical factors that could underlie differences among CA3PCs. Resting $V_m$ and $R_{in}$ weakly correlated with CSB rate ($V_{rest}$ vs. 300 pA: $p = 0.028$, Spearman $R = -0.200$; $V_{rest}$ vs. 600 pA: $p = 0.073$, Spearman $R = -0.163$, $n = 120$ cells; $R_{in}$ vs. 300 pA: $p = 0.061$, Spearman $R = 0.172$, $R_{in}$ vs. 600 pA: $p = 0.073$, Spearman $R = 0.165$, $n = 118$ cells, Supplementary Fig. 5a, b), indicating that these basic electrophysiological properties are unlikely to simply account for the heterogeneity. Therefore, we examined the involvement of VDCs. In particular, we focused on two channel types that are expressed differently along the proximodistal axis of CA3 and are typically localized to somatodendritic compartments of PCs: HCN and Kv2.2 channels.

The h-current ($I_h$) is a major regulator of dendritic excitability and integration in several PC types, where it is expressed mainly in dendrites[41]. Expression of HCN subunits (mediating $I_h$) and the hyperpolarization-activated sag is weaker in CA3 than in CA1 PCs[33,41,42], however, the subcellular channel localization in CA3PCs has not yet been investigated. A decreasing gradient of $I_h$-mediated sag from CA2 to CA3c has been reported in mouse

hippocampal slices[33]. Therefore we first investigated the impact of $I_h$ on CSB propensity.

We confirmed a strong spatial gradient of $I_h$: the sag ratio (SR; see Methods section) was higher in CA3PCs than that in CA1PCs (Fig. 6a, b, $n = 109$ CA3PCs, $n = 12$ CA1PCs, $p < 0.001$, Mann–Whitney test), and increased from distal (CA3a) to proximal (CA3c) direction (Fig. 6c, Spearman $R = -0.505$, $p < 0.001$, $n = 109$ cells). Within CA3a-b, where the sag was more substantial, we found that cells with larger sag (lower SR) had higher CSB rate (Fig. 6d, 300 pA: $p < 0.001$, Spearman $R = -0.472$, $n = 51$ cells; 600 pA: $p < 0.0001$, Spearman $R = -0.509$, $n = 49$ cells). Similar observations were reported in other cell types[43,44], possibly reflecting coregulation of $I_h$ expression with other conductances involved in burst generation. In contrast to CA3a-b, in CA3c no correlation was found between the SR and CSB rate (Supplementary Fig. 6a, b, 300 pA: $p = 0.130$, Spearman $R = -0.231$, $n = 44$ cells; 600 pA: $p = 0.292$, Spearman $R = -0.162$, $n = 44$ cells).

To investigate the role of $I_h$ in regulating CSBs, we bath-applied the $I_h$ inhibitor ZD7288 (10 μM), and compared CSB rate measured during a 20–30-min-long stable baseline period (ctr) and 10–20 min after $I_h$ block. Control experiments showed that CSB rate was stable without intervention (Supplementary Fig. 6c). In PCs in CA3a-b, ZD7288 induced hyperpolarization (compensated by appropriate current injection) and eliminated the sag ($p = 0.027$, $n = 6$, Wilcoxon test, Fig. 6e, f). Blockade of $I_h$ led to an increase in CSB rate by 600 pA $I_{inj}$ (measured with baseline $V_m$ readjusted to −70 mV; Fig. 6g, h, $n = 6$, 300 pA: $p = 0.224$; 600 pA: $p = 0.027$, Wilcoxon test). However, the relationship between control SR and CSB rate after ZD application remained similar to that under control conditions (Fig. 6i), suggesting that $I_h$, although does limit CSB generation, is not primarily responsible for the heterogeneity of CSB propensity within CA3a-b. Application of ZD7288 in PCs in CA3c—while it eliminated the small sag (Fig. 6e)—did not increase CSB rate (Fig. 6j, $n = 5$; $p = 0.179$ for both $I_{inj}$ levels, Wilcoxon tests). ZD7288 altogether had a significantly smaller effect on CSB rate in CA3c than in CA3a-b (Fig. 6k; two-way ANOVA, $p = 0.002$ for subregion, $p = 0.021$ for $I_{inj}$ level, $p = 0.016$ for interaction), indicating that the role of $I_h$ in CSB regulation is minor in CA3c.

Dendrites of PCs express various types of $K^+$ currents, which may also influence CSB propensity. Interestingly, recent studies revealed selective expression of Kv2.2 $K^+$ channel subunits in PCs of the CA3c subregion, while the other family member, Kv2.1 is expressed relatively homogeneously in CA3PCs[45,46], predicting increased total Kv2 channel-mediated currents in CA3c. Kv2 channels mediate delayed rectifying voltage-gated $K^+$ current[45–47], and in other PC types are typically expressed in somatodendritic compartments.

Immunolabelling for Kv2.1 and Kv2.2 subunits confirmed the proximodistally decreasing gradient of Kv2.2 (but not Kv2.1) expression, and showed that Kv2.2 is localized in somata and

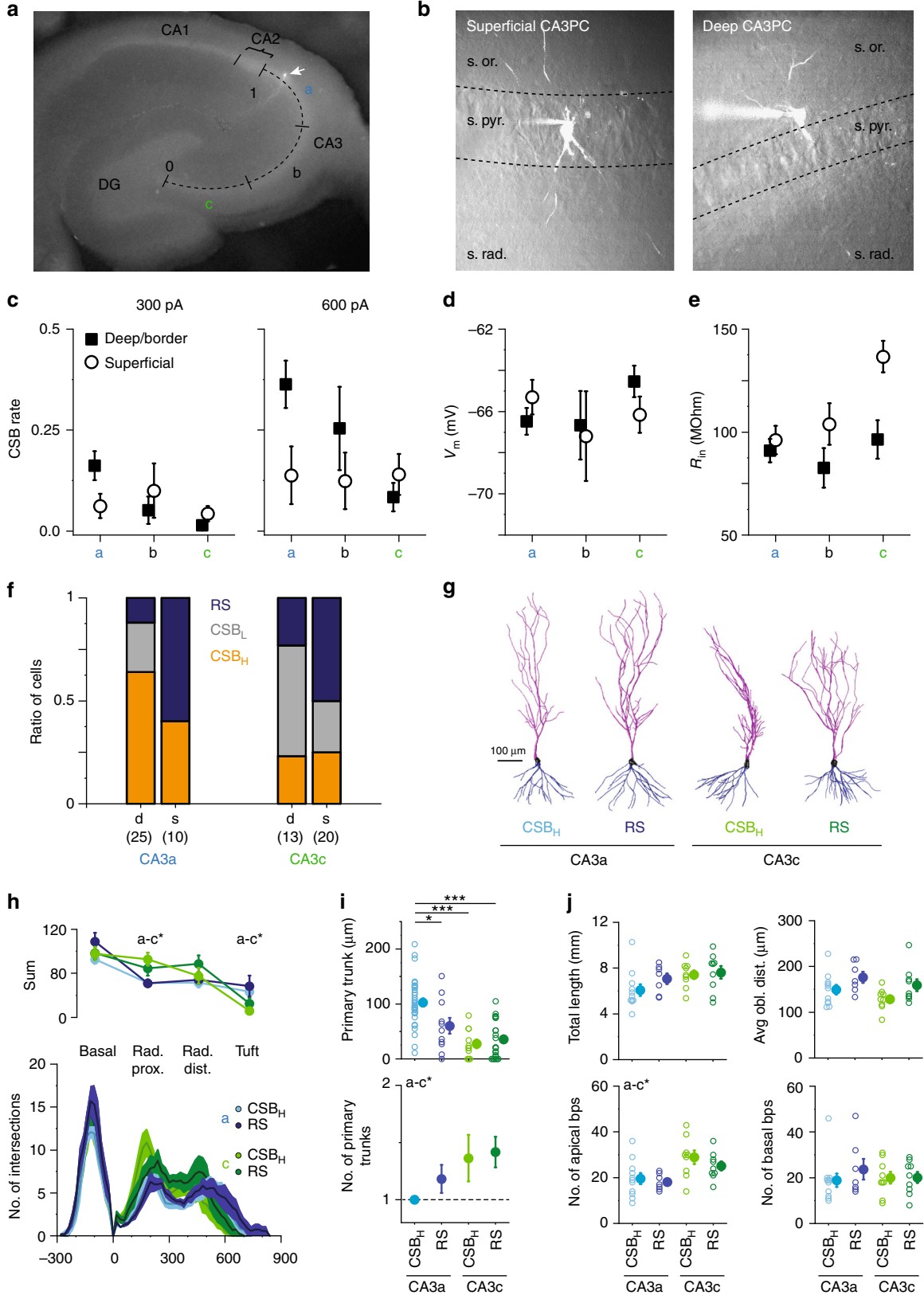

proximal dendrites of CA3cPCs, but is hardly detectable in CA3aPCs (Fig. 7a–g). This suggests that Kv2 channels may limit CSB propensity of PCs in CA3c stronger compared to CA3a-b. We therefore tested the effect of the Kv2 channel inhibitor guangxitoxin-1E (GxTx, 0.1 μM, 10–20 min bath application) on

CSB propensity in these different subregions. Application of GxTx strongly increased CSB rate in CA3c by 300 and 600 pA $I_{inj}$, in both RS and CSB cells (Fig. 7h–j, $n = 6$, 300 pA: $p = 0.027$; 600 pA: $p = 0.027$, Wilcoxon test), even though it did not modify basic electrophysiological properties such as $V_m$, $R_{in}$ (Fig. 7i) or

**Fig. 5** Cellular location dependence of CSB propensity. **a** Definition of proximodistal CA3 subregions. **b** Definition of radial CA3PC groups. Superficial cells had soma well within the borders of str. pyramidale. Deep/border cells had soma on, or outside of, the str. pyramidale-str. oriens border. Layer border was determined by either Dodt contrast or 2P image stacks. **c** CSB propensity to 300 (left) and 600 (right) pA $I_{inj}$ series in the six CA3PC groups. **d**, **e** Resting $V_m$ (**d**) and $R_{in}$ (**e**) of the six CA3PC groups. Symbols and error bars in **c**–**e** represent mean ± s.e.m. **f** Ratio of CSB$_H$, CSB$_L$, and RS CA3PCs in CA3a and c. Number of deep and superficial cells, respectively: CA3a: 25/10, CA3b: 7/5, CA3c: 13/20. **g** Example CA3PC morphologies. **h** Sholl analysis on reconstructed CSB$_H$ and RS cells in CA3a and CA3c. Upper panel: sum of dendritic intersections in str. oriens, proximal str.radiatum (100–280 µm), distal str.radiatum (300–500 µm) and SLM (>520 µm). Symbols and error bars represent mean ± s.e.m. The branching profile was different between CA3a and CA3c PCs, but did not depend on CSB propensity (two-way repeated measures ANOVA, $p < 0.001$ for subregion, $p = 0.310$ for bursting phenotype, $p = 0.409$ for interaction). In lower panel, lines and shadings represent mean ± s.e.m., respectively. **i** Primary trunk length and number, measured based on 2P image stacks ($n = 34$ for CA3a CSB$_H$, $n = 11$ for CA3a RS, $n = 11$ for CA3c CSB$_H$, $n = 24$ for CA3c RS). **j** Total dendrite length, average distance of oblique dendrites from soma, and branch point numbers, as measured in reconstructed neurons ($n = 11$ for CA3a CSB$_H$, $n = 7$ for CA3a RS, $n = 9$ for CA3c CSB$_H$, $n = 9$ for CA3c RS). Two-way ANOVA with Holm-Bonferroni correction. Filled symbols and error bars in **i**–**j** represent mean ± s.e.m. *$p < 0.05$; ***$p < 0.001$. Source data are provided as a Source Data file

properties of the first AP (Fig. 7k). On the other hand, although GxTx modestly increased CSB rate in PCs in CA3a-b ($n = 7$, 300 pA: $p = 0.043$; 600 pA: $p = 0.027$, Wilcoxon test, Fig. 7l and Supplementary Fig. 6d), its effect in these subregions was smaller than that in CA3c (Fig. 7m, two-way ANOVA, $p < 0.001$ for subregion, $p = 0.012$ for $I_{inj}$ level, $p = 0.282$ for interaction). In summary, generation of CSBs is under the control of specific ion channels in different subregions of CA3.

## Discussion
Previous studies in CA1 and L5 PCs led to an influential model of how active dendrites of cortical PCs integrate and transform synaptic inputs to specific AP output[48–50]. According to the model, in thin (basal, oblique, or tuft) dendrites, spatiotemporally correlated synaptic input can produce local $Na^+$ and/or NMDA spikes. Robust multibranch input in the apical dendritic arbor can also trigger $Ca^{2+}$ spikes, produced in the apical trunk. These $Ca^{2+}$ spikes, together with NMDARs, generate a prolonged dendritic plateau potential that produces slow afterdepolarization at the soma, evoking a characteristic CSB. In both CA1 and L5 PCs, the most efficient excitatory pattern to evoke dendritic plateaus and CSBs is the conjunction of proximal (mostly by local afferents) and distal apical input (mostly by long-range or feedback afferents)[6,13,50]. Thereby, dendritic plateaus and CSBs were proposed as an associative device, coupling external information of the world with its internal representation[50], and were suggested as an important component for biological implementation of error-driven learning in multi-layer networks[51].

We examined whether the generation of CSBs follows the above integration scheme in CA3PCs. Our results uncover both similarities and differences from the canonical model. On one hand, we confirm that dendritic (mainly $Ca^{2+}$) spikes underlie CSBs in CA3PCs as well. On the other hand, we found that the ability to generate CSBs, and the input patterns triggering these responses are not uniform among CA3PCs. While a subgroup of CA3PCs (RS cells) produces CSBs only upon combined proximo-distal dendritic depolarization similar to the canonical model, other CA3PCs (CSB$_H$ cells) can produce CSBs even to correlated inputs localized to an isolated dendritic segment. This means that CSB$_H$ cells can respond with CSBs to a single presynaptic input type (e.g., RC or EC). Therefore, in these cells $Ca^{2+}$ spikes do not necessarily represent an associative signal between different synaptic information types and the role of CSBs is unlikely to mediate an error-driven learning signal. If the heterogeneity of dendritic integration applies also under in vivo conditions, our results imply that caution should be taken to make assumptions about the afferent input pattern evoking CSBs in individual CA3PCs.

CSB propensity varied along the proximodistal CA3 axis so that CSB rate was higher in distal CA3PCs, consistent with

previous reports on CA3PC burst spiking[15,28,36,37] (although those studies did not specifically investigate CSBs). We identified two ion channels—HCN and Kv2 channels—that may contribute to this gradient. We measured stronger $I_h$-mediated sag in distal than in proximal CA3PCs in line with previous data[33], and (consistent with a dampening impact of $I_h$ on dendritic excitability) $I_h$ blockade increased CSB propensity stronger in distal CA3. Interestingly, the sag ratio and CSB rate were inversely related in distal CA3PCs: the more $I_h$, the higher the CSB rate was. This counterintuitive correlation suggests a more complex mechanism, perhaps coexpression of $I_h$ with other ion channel(s) involved in CSB regulation, or compensatory upregulation of $I_h$ in bursting cells. Furthermore, weaker sag in CA3c did not result in higher CSB propensity, indicating that other mechanisms regulate CSB propensity in the proximal subregion. Indeed, we pinpointed another factor, currents mediated by Kv2 channels, that regulates CSBs more strongly in CA3c. Kv2.2 channels are almost exclusively expressed in CA3c, whereas Kv2.1 expression is uniform in CA3. Localized primarily in somatic and proximal apical dendritic membrane, Kv2 channels appear to be well-suited to regulate initiation and/or coupling of dendritic and somatic spike generation. Thus, differential expression of ion channels, along with gradients of synaptic strength and connectivity[33], may allow for proximodistal subregion-selective regulation of CSBs.

CSB propensity was also related to radial position, with cells in deep layers of distal CA3 having the highest propensity to burst. As early-born CA3PCs are preferentially distributed in deep CA3a-b and spike more in bursts[36], embrionic origin may partly account for different phenotypes of CA3PCs. A recent report[37] in juvenile animals also found higher burst rate in PCs in deep layers of CA3a-b. While the authors reported that these bursting cells lacked thorny excrescences[37], our deep distal CSB$_H$ cells did possess complex spine-like postsynaptic structure(s). Thus, although athorny PCs may exist in the adult rat CA3, the majority of even the thorny deep distal CA3PCs displays complex spike bursting phenotype. Nonetheless, it remains plausible that these cells may comprise a morpho-functionally distinct, perhaps parallel processing layer, as proposed by Hunt et al.[37].

Heterogeneity among CA3PCs was still substantial in each topographical category, suggesting that other factors also contribute to the cell-to-cell variability to produce CSBs. These most likely include passive and active dendritic properties. While we did not find major general dendritic morphological signatures of variable CSB propensity, more subtle trends may exist that were not evident from analysis of our limited reconstruction dataset. Furthermore, dendritic diameter (which we did not investigate) may be an important factor. In addition to morphology, differential density and/or spatial pattern of dendritic VDCs, especially VGCCs and various $K^+$ channel types that we did not examine here (e.g., A-type[22] or inward rectifiers[23]) could contribute to

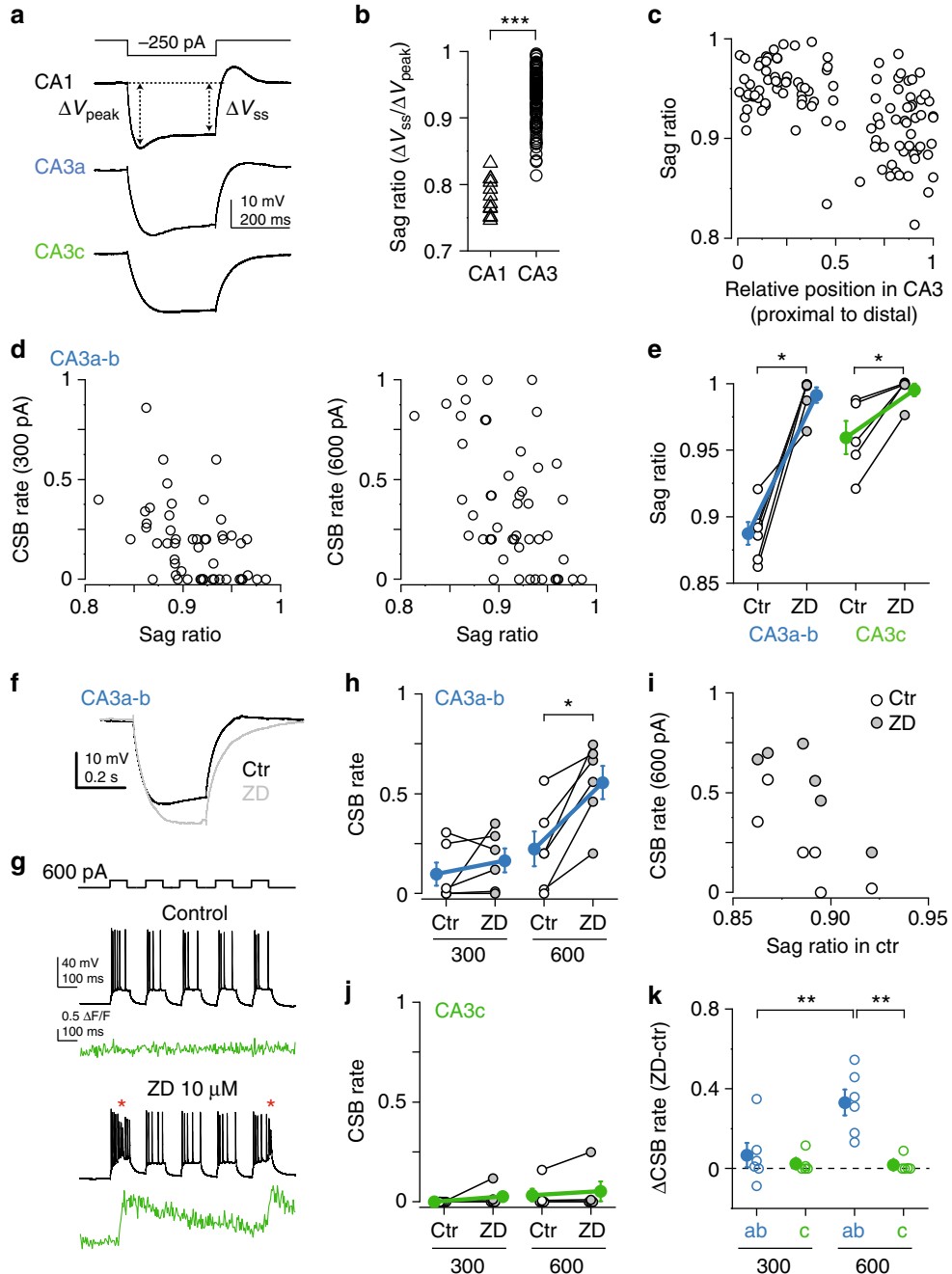

**Fig. 6** Role of $I_h$ in regulation of CSB propensity. **a** Representative voltage responses to hyperpolarization in a CA1, CA3a and CA3c PC. **b** Sag ratio (SR) in CA1 and CA3 PCs. **c** Proximodistal gradient of SR within CA3. **d** Correlation between SR and CSB rate to 300 pA (left) and 600 pA (right) $I_{inj}$ in CA3a-b. **e** Effect of 10 μM ZD7288 on the SR. **f** Sag is eliminated by ZD7288 (representative recording from a CA3aPC). Hyperpolarization by ZD7288 was compensated by constant current injection. **g** Representative experiment testing the effect of ZD7288 on CSB rate. CSBs are indicated by red asterisks. **h** Summary of the effect of ZD7288 on CSB rate by 300 and 600 pA $I_{inj}$. **i** Relationship between initial SR and CSB rate in control condition and in ZD7288. **j** Same as **h** but in CA3c. **k** CSB rate difference induced by ZD7288 in CA3a-b and CA3c. Filled symbols and error bars in **e**, **h**, **j**, and **k** represent mean ± s.e.m. *$p < 0.05$; CA3PCs with distinct CSB propensity **$p < 0.01$; ***$p < 0.001$. Source data are provided as a Source Data file

the observed heterogeneity. In fact, as CSBs result from an interplay of somatic and dendritic electrogenesis, any conductance that affects the properties of APs, $Ca^{2+}$ spikes and/or electrotonic spread of depolarization may influence CSB propensity. Since many ion channels are regulated in an activity-dependent manner, the functional heterogeneity of CA3PCs we observed may also reflect different preceding in vivo firing activity and activity-dependent plasticity of dendritic excitability[52–54]. Alternatively, the observed heterogeneity could be the result of

homeostatic regulation of excitability, which may contribute to determining which CA3PCs can be preferentially recruited into a new engram.

Our results and previous work indicate that dendrites of CA3PCs can efficiently produce both local ($Na^+$[22,23], NMDA[23,24]) and widespread ($Ca^{2+}$) d-spikes upon correlated input. What may be the functional roles of d-spikes in CA3 circuit computations, and how may cell-to-cell variability fit into these functions? The CA3 network is thought to enable both

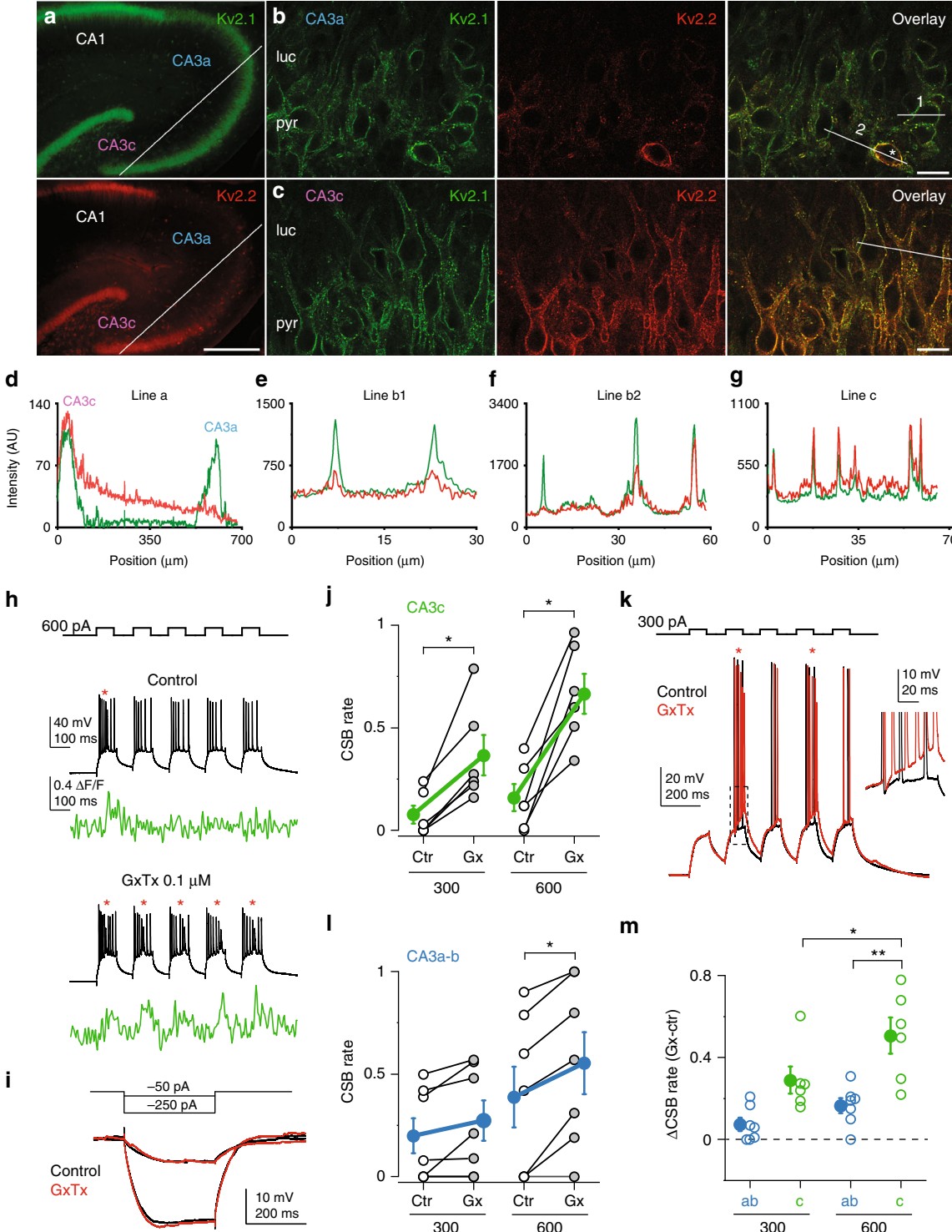

**Fig. 7** Role of Kv2 channels in regulating CSB propensity. **a** Double immunofluorescent reaction showing the distribution of Kv2.1 (top) and Kv2.2 (bottom) subunits in the hippocampus. **b, c** High magnification co-localization of the Kv2.1 and Kv2.2 subunits in the CA3a (**b**) and CA3c (**c**) areas. Asterisk in **b** marks an interneuron intensely labeled for the Kv2.2 subunit in CA3a. Scale bars: 200 μm (**a**), 20 μm (**b**, **c**). The stainings were replicated 19 times in 10 rats. **d–g** Intensity profiles of the Kv2.1 (green) and Kv2.2 (red) immunosignals measured along the indicated white lines in **a–c**. AU, arbitrary unit. **h** Representative experiment testing the effect of GxTx (0.1 μM) on CSB rate in a CA3c PC. CSBs are indicated by red asterisks. Low-affinity $Ca^{2+}$ dye OGB-6 was used in this experiment. **i** Representative voltage responses to hyperpolarization in a CA3c PC in control and after GxTx application. **j** Summary of the effect of GxTx on CSB rate by 300 and 600 pA $I_{inj}$ in CA3c. **k** Overlaid representative voltage traces before (control) and after wash-in of GxTx in a CA3c PC. **l** Summary of the effect of GxTx on CSB rate by 300 and 600 pA $I_{inj}$ in CA3a-b. **m** Difference in CSB rate induced by GxTx in CA3a-b and CA3c. Filled symbols and error bars in **j**, **l**, and **m** represent mean ± s.e.m. *$p < 0.05$; **$p < 0.01$. Source data are provided as a Source Data file

encoding of novel context-specific external input patterns into orthogonal ensemble representations (i.e., pattern separation), as well as complete retrieval of these representations from a part during memory recall (i.e., pattern completion)[55–58]. These CA3 computations are classically explained by connectivity and plasticity of synaptic inputs. Formation of new CA3PC ensembles in a novel environment is thought to be governed by MFs, which discharge a sparse set of CA3PCs, instructing plasticity of coactive RC and EC synapses; the strengthened connections could then recruit the entire ensemble from a part in a similar but not identical context[55–58]. Although these theories ignore nonlinear input interactions, recent models proposed that nonlinear dendrites may enrich the computational functions of CA3PCs[59,60]. During encoding of novel information (pattern separation), d-spikes may facilitate initial AP firing of, and promote synaptic plasticity among coactive CA3PCs—perhaps specifically those that receive associated MF and RC/EC input[60,61]. During memory recall (pattern completion), d-spikes may amplify the response to degraded but sufficiently strong RC and/ or EC input, ensuring reliable AP firing by ensemble members. This latter process may be further promoted by activity-dependent dendritic plasticity, leading to increased dendritic excitability in ensemble members. Among d-spike types, $Ca^{2+}$ spikes can be particularly powerful, not only by enhancing burst firing but also by producing large widespread dendritic $Ca^{2+}$ signal that may induce rapid synaptic plasticity, as proposed in CA1PCs[9].

The proximodistal differences reported in connectivity and ensemble dynamics[15,31,32] suggest that CA3 should not be considered as a single associative network. Instead, CA3c was proposed to cooperate with DG to perform pattern separation, whereas the more interconnected CA3a-b may be more involved in pattern completion and generation of SWRs[62]. In this context, the generally higher ability of distal CA3PCs to fire CSBs (even to correlated RC input alone) may help to efficiently recruit these neurons and their downstream targets during reactivation processes, with $CSB_H$ neurons particularly suitable to initiate SWRs[37]. In contrast, the generally lower CSB propensity of CA3c cells may render CSBs to represent dual-pathway (proximal MF or RC, and distal EC) multiplicative integration and contribute to selection of more orthogonal representations by CA3PCs that receive both perisomatic and coincident distal input. Thus, the different channel assortment of proximal and distal CA3PCs regulating integrative and firing properties may open the possibility to modulate pattern separation and completion selectively. Alternatively, microcircuits of CA3PCs with distinct CSB propensity along CA3 could be dedicated to perform parallel processing of information[37,63]. Whether intrinsic excitability plays an instructive role in feature-selective activity of CA3PCs similar to other neurons[6,10,11,64–66], and how d-spikes can be incorporated into the input-based model of associative functions of CA3 remains to be elucidated.

## Methods

**Hippocampal slice preparation.** Adult male Wistar rats (7–12-week-old) were used to prepare 400-μm-thick slices from the hippocampus[23], according to methods approved by the Animal Care and Use Committee of the Institute of Experimental Medicine of the Hungarian Academy of Sciences, and in accordance with the Institutional Ethical Codex, Hungarian Act of Animal Care and Experimentation (1998, XXVIII, section 243/1998), and European Union guidelines (86/609/EEC/2 and 2010/63/EU Directives). Animals were deeply anaesthetized with 5% isoflurane and quickly perfused through the heart with ice-cold cutting solution containing (in mM): sucrose 220, NaHCO$_3$ 28, KCl 2.5, NaH$_2$PO$_4$ 1.25, CaCl$_2$ 0.5, MgCl$_2$ 7, glucose 7, Na-pyruvate 3, and ascorbic acid 1, saturated with 95% O$_2$ and 5% CO$_2$. The brain was quickly removed and slices were prepared in cutting solution using a vibratome (Vibratome, St. Louis, MO, or Leica VT1000A, Leica Biosystems GmbH, Nussloch, Germany). Slices were incubated in a submerged

holding chamber in ACSF at 35 °C for 30 min and then stored in the same chamber at room temperature.

**Patch-clamp recordings.** For recording, slices were transferred to a custom-made submerged recording chamber under the microscope where experiments were performed at 32–34 °C in ACSF containing (in mM): NaCl 125, KCl 3, NaHCO$_3$ 25, NaH$_2$PO$_4$ 1.25, CaCl$_2$ 1.3, MgCl$_2$ 1, glucose 25, Na-pyruvate 3, and ascorbic acid 1, saturated with 95% O$_2$ and 5 % CO$_2$ (see ref. [23]). The cells were visualized using Zeiss Axio Examiner or Olympus BX-61 epifluorescent microscope under infrared illumination and water immersion lens (63× or 60× during recording, 20× or 10× for z-stack, Zeiss or Olympus). To access neurons with mostly preserved dendritic arborization, in every slice we selected a region within CA3 where the apical trunks of most neurons were oriented slightly downwards (directed into the tissue), and we targeted somata located >50 μm deep in the slice. Current-clamp whole-cell recordings from the somata of hippocampal CA3 (or in some experiments CA1) pyramidal neurons were performed using BVC-700 amplifier (Dagan, Minneapolis, MN) in the active "bridge" mode, filtered at 3 kHz and digitized at 50 kHz. Patch pipettes (2–6 MΩ) were filled with a solution containing (in mM): K-gluconate 134, KCl 6, HEPES 10, NaCl 4, Mg$_2$ATP 4, Tris$_2$GTP 0.3, phosphocreatine 14 (pH = 7.25), typically complemented with 50 μM Alexa Fluor 594 and 100 μM Oregon Green BAPTA-1 (OGB-1) or Oregon Green BAPTA-6 (OGB-6) (all fluorescent dyes from Invitrogen-Molecular Probes). While we mostly used OGB-1, we had two reasons to perform experiments also with the low affinity indicator OGB-6F: first, in some CA3PCs the large CSB-associated dendritic Ca$^{2+}$ signals caused saturation; second, we wished to examine the possibility that Ca$^{2+}$ buffering by the high affinity OGB-1 may influence the CSB rate per se for example by limiting the recruitment of Ca$^{2+}$-activated K$^+$ channels. In some experiments the pipette solution included ~0.1–0.3 % biocytin (Sigma), or 100 μM Alexa Fluor 488 instead of the above fluorescent dyes. We found similar burst propensity of CA3PCs using the various fluorescent dyes with different Ca$^{2+}$ affinities (OGB-1 ($n = 81$) vs. OGB-6 ($n = 37$) at 300 pA: $p = 0.190$, 600 pA: $p = 0.812$, Mann–Whitney test) or Ca$^{2+}$-insensitive Alexa 488 ($n = 5$), making it unlikely that different levels of Ca$^{2+}$ buffering influenced CSB propensity, and therefore results obtained with different Ca$^{2+}$-sensitive dyes were pooled. Membrane potentials are reported without correction for liquid junction potential (10 mV). Series resistance was typically 10–20 MΩ, frequently checked and compensated with bridge balance and capacitance compensation; recordings were terminated when series resistance exceeded 35 MΩ. Only CA3PCs with initial resting membrane potential ($V_m$) more negative than −60 mV after break-in were used for experiments. Cells were usually kept at −68–72 mV with appropriate constant current injection, unless otherwise indicated. CA1 PCs were held at ~−65 mV unless otherwise indicated. Typically one neuron per slice was recorded.

After establishing whole cell current clamp configuration, cells were loaded for >20 min to allow sufficient diffusion of the fluorophores to the dendritic tree. All CA3 neurons were first carefully inspected and confirmed (1) to have thorny excrescences on thick parent dendrites as well as small spines on thinner dendritic branches, and (2) that no main trunk was cut. We next measured CSB rate in response to a series of somatic depolarizing current injections ($I_{inj}$, five 100-ms-long pulses of 300–600 pA with 80.55-ms-long interpulse intervals) from ~−70 mV baseline $V_m$. In a subset of CSB$_H$ cells we also applied 200 pA $I_{inj}$ (7 of 15 cells fired CSBs at 200 pA). After determining their CSB propensity, neurons were assigned to different experiments (e.g., pharmacological manipulations, synaptic stimulation).

Dendritic patch-clamp recordings were established at ~100–200 μm distance from the soma (in proximal str. radiatum, $n = 5$ in CA3a-b, $n = 5$ in CA3c) usually on a secondary trunk. Series resistance was <50 MOhm. Dendritic $V_m$ ranged −57 to −73 mV (−64.5 ± 1.7 mV), $R_{in}$ was 99 ± 11 MOhm ($n = 10$).

**Two photon imaging and uncaging.** A dual galvanometer based two photon scanning system (Prairie Technologies, Middleton, WI, USA) was used to image the patched neurons and to uncage glutamate at individual dendritic spines similarly to published methods[23,52,53,67]. Two ultrafast pulsed laser beams (Chameleon Ultra II; Coherent, Auburn, CA, USA) were used: one laser at 920 or 860 nm for imaging OGB and Alexa Fluor dyes, respectively, and the other laser tuned to 720 nm to photolyze MNI-caged-L-glutamate (Tocris; 10 mM in ACSF) that was applied through a puffer pipette with a ~20–30-μm-diameter, downward-tilted aperture above the slice using a pneumatic ejection system (PDES-02TX (NPI, Tamm, Germany).

The intensity of the laser beams was independently controlled with electro-optical modulators (Model 350–80, Conoptics, Danbury, CT, USA). Linescan Ca$^{2+}$ measurements were performed at ~200–300 Hz in distal dendritic segments, located in SLM or near the str. radiatum-SLM border. Glutamate uncaging was performed at a clustered set of spines on a single dendritic segment, using 0.3–0.5 ms uncaging duration with 0.1 ms intervals between synapses (quasi-synchronous stimulation), repeated 5 times at 40–50 Hz (i.e., a gamma burst stimulus). Uncaging laser power was adjusted to yield compound somatic voltage responses peaking near the threshold of APs, so that both subthreshold and suprathreshold responses could be evoked. In cases when more than one dendrite of the same category (basal, oblique or tuft) was tested in the same cell, the data were averaged.

**Focal electrical stimulation**. Focal electrical synaptic stimulation (BioStim stimulator system, Supertech Ltd., Pecs, Hungary) was performed using silver wires inserted into a theta pipette with a slightly broken tip, filled with ACSF containing 50 µM Alexa 594. The tip of the theta pipette was placed into SLM, guided by the fluorescent image of the dendrite overlaid with two-photon Dodt image visualizing the theta pipette. In some slices, electrical stimulation in the presence of GABAR blockers triggered population bursts; since the population bursts made it impossible to evaluate CSBs, these experiments were excluded from the analysis.

**Chemicals**. D-AP5, NBQX disodium salt, tetrodotoxin, SR95531 hydrobromide, CGP55845, guangxitoxin, ZD7288 (all from Tocris) were dissolved in distilled water in stock solutions, aliquots were stored at −20 °C and used on the day of experiment. $CdCl_2$ was purchased from Sigma-Aldrich. All inhibitors were applied in the bath. Note that the effective concentration of D-AP5 in the uncaging experiments is expected to be reduced at the stimulated dendritic region during puffing of drug-free MNI-glutamate solution.

**Fluorescent immunohistochemistry**. Immunofluorescent reactions were performed on free floating 70 µm thick hippocampal Vibratome sections obtained either from perfusion fixed brains (Fig. 7a) or immersion fixed acute hippocampal slices (Fig. 7b, c). We observed intense and identical labeling patterns with both fixation methods. For perfusion fixation, two adult rats were anesthetized and perfused through the aorta with 4% paraformaldehyde and 15 v/v% picric acid in 0.1 M PB (pH = 7.3) for 15 min. For immersion fixed slices, acute hippocampal slices (cut and used as described for acute hippocampal slice preparation and patch-clamp recordings, $n = 17$ slices from 8 rats) previously incubated in ACSF were transferred to a fixative containing 2% paraformaldehyde and 15 v/v% picric acid in 0.1 M PB (pH = 7.3) for 2 h at room temperature or overnight at 4 °C, then washed in PB and re-sectioned to 70–100 µm thick sections. Following a blocking step in 10% normal goat serum (NGS) made up in Tris-buffered saline (TBS, pH = 7.4), the sections were incubated in the mixture of mouse monoclonal anti-Kv2.1 (catalog 75–315, clone L80/21, RRID: AB_2315863, IgG3, 1:500, UC Davis/NIH NeuroMab Facility, Davis, CA) and Kv2.2 (catalog 75–369, clone 372B/1, RRID: AB_2315870, IgG1, 1:500, Neuromab) antibodies diluted in TBS containing 2% NGS and 0.2% Triton X-100. Alexa488 conjugated goat anti-mouse IgG3 (1:500; Jackson ImmunoResearch, West Grove, PA) and Cy3 conjugated goat anti-mouse IgG1 (1:500, Jackson ImmunoResearch) IgG-subclass-specific secondary antibodies were used to visualize the immunoreactions. Specificity of the immunolabelings with the same Kv2.1 and Kv2.2 antibodies was previously confirmed in knockout mice[45].

Low magnification fluorescent images were aquired with an Olympus BX-62 epifluorescent microscope at 10x magnification, whereas high magnification fluorescent images were taken with an Olympus FV1000 confocal microscope using a 60x objective (NA = 1. 35). Line scan analysis on the fluorescent images was performed with Fiji.

**Data analysis**. Analysis of voltage and $Ca^{2+}$ recordings was performed using custom-written macros in IgorPro (WaveMetrics, Lake Oswego, OR, USA). $Ca^{2+}$ signal amplitude was determined as the maximum average of 5 consecutive points, measured from the 20-ms baseline preceding the $I_{inj}$ pulse.

CSBs evoked by $I_{inj}$ were identified based on the combination of the following properties: (1) ≥2 high-frequency APs with progressively decreasing peak amplitude, preceded by at least one additional simple AP in the given $I_{inj}$ pulse, (2) riding on an underlying slow afterdepolarization that followed the first AP of the CSB, and (3) typically accompanied by a $Ca^{2+}$ signal in a distal apical dendrite. The first two criteria were mandatory, whereas the distal dendritic $Ca^{2+}$ signal was used as a complementary parameter, because in some cases saturation of OGB-1 during sequential CSBs, or relatively poor signal-to-noise with OGB-6 prevented reliable evaluation of the $Ca^{2+}$ signal. CSBs evoked by $I_{inj}$ were usually following or embedded into simple AP firing. We note that AP doublets or series with short inter-spike intervals did not necessarily meet the above criteria for CSBs. The presence of the slow depolarization was evaluated based on the subthreshold $V_m$ trace generated by removal of APs and linear interpolation of the gap, and the amplitude of the slow depolarization on a given $I_{inj}$ pulse was calculated as the $V_m$ difference between the threshold of the first AP and the peak of the subthreshold trace measured in that pulse. CSBs were in most cases clearly detectable by visual inspection of the voltage and $Ca^{2+}$ traces. Due to the substantial variability in the properties of CSBs as well as in the AP firing rate and adaptation pattern across CA3PCs, a uniform set of numerical parameters could not be used for CSB detection in all cells; nevertheless, these properties were characteritic in a given cell. In a subset of cells ($n = 24$), visual identification was confirmed by calculating the product of three electrophysiological parameters (1. amplitude of the slow depolarization, 2. maximum relative increase of AP frequency and 3. maximum decrease of peak AP amplitude from that of the first AP), which reliably separated pulses with and without CSBs in agreement with visual identification.

In uncaging experiments (where, due to the different stimulation pattern evoking gradual depolarization, the CSBs were not embedded in AP activity) CSBs were identified as (1) a high-frequency burst of ≥3 APs with decreasing amplitude (2) riding on a slow ADP. In uncaging experiments $Ca^{2+}$ signals were not used,

because NMDAR activation by the strong synaptic stimulation induced large local $Ca^{2+}$ elevation per se[23].

CSB propensity was quantified by calculating the CSB rate, which is the relative frequency spanning the range 0–1. In the $I_{inj}$ experiments, CSB rate was calculated by dividing the number of $I_{inj}$ pulses displaying CSB with the total number of $I_{inj}$ pulses (5 per trace, usually from 10 repetitions, i.e., 50 pulses in total). In some cells 300 pA $I_{inj}$ evoked no APs yet or only on some of the pulses; nevertheless, these data were also included in the analysis (as pulse with no CSBs). In electrical stimulation experiments, CSB rate was calculated by dividing the number of synaptic bursts evoking CSB with the total number of synaptic bursts (5 per trace, usually from 10 repetitions, i.e., 50 synaptic bursts in total, see Fig. 4f). In uncaging experiments, CSB rate was calculated by dividing the number of traces (with a single gamma burst stimulus, see Fig. 4b) evoking CSB with the total number of traces evoking at least 1 AP (usually 5–9 repetitions). We note that while in this study we focused on the propensity (i.e., relative frequency) of CSBs, the properties of the CSBs (i.e., duration, kinetics, number of APs involved) also showed large variation across CA3PCs.

Resting $V_m$ was registered as the voltage displayed by the amplifier immediately after break-in and transfer to current clamp configuration. Input resistance was determined at the end of voltage responses to 50–100 pA, 300 ms hyperpolarizing step current injections from ~−70 mV baseline $V_m$. The sag ratio (SR) was calculated by dividing the steady-state voltage response by the peak voltage response to −250 pA hyperpolarizing step current injections from ~70 mV baseline $V_m$. In a few cells with very high rate of spontaneous EPSPs the above electrophysiological properties were not determined. AP threshold was measured as the voltage value where $dV/dt$ exceeded 20 V s$^{-1}$. AP onset time was measured as the latency of the first AP threshold in response to 600 pA step current injection.

The afterdepolarization of an individual AP (ADP$_{AP}$, Fig. 3i) during the CSB was measured as the maximum subthreshold membrane potential between 4 and 8 ms after the AP peak relative to the AP threshold. Traces using $I_{inj}$ levels at or 100 pA above the lowest $I_{inj}$ evoking CSBs were used, and maximal ADP$_{AP}$ values measured in multiple CSBs were averaged. APs occurring <20 ms after the start of the $I_{inj}$ pulse were excluded from the analysis. Measurements from dendritic recordings were compared with results from interleaved somatic recordings from other CA3PCs performed during the same time period.

**Morphological analysis**. Alexa Fluor 594 fluorescence or biocytin labeling was used for post hoc verification of the localization of neurons along the proximodistal axis of CA3 and morphological analysis.

The relative proximodistal position was calculated based on measurements on low-magnification fluorescent images of the whole hippocampus. The distal border of CA3a (from CA2) was considered to be located at 200 µm from the sudden flaring of the pyramidal cell layer (i.e., CA1–CA2 border, see Fig. 5a). The border of str. pyramidale and str. oriens was determined on either 2P Dodt contrast or 2P fluorescent image stacks showing the local population of pyramidal cells. A PC was considered to be superficial if its soma lay completely in str. pyramidale, not touching the border with str. oriens.

Detailed dendritic morphological and distance measurements were performed on stacked images of dye-loaded neurons (collected at the end of the experiment with a 63× or 20× objective) using ImageJ (NIH, Bethesda, MD, USA). Primary apical trunk length and number (Fig. 5i) were measured directly on the collapsed 2P fluorescent stacks in ImageJ by manually drawing a segmented line from the apical edge of the soma to the first main bifurcation. Other parameters characterizing the complexity of the dendritic tree (Fig. 5j) were extracted from semiautomated analysis of fully reconstructed CA3PCs with the most complete dendritic arbor using Vaa3D software (www.vaa3d.org, see ref. [68,69]), followed by visual inspection, and manual corrections where necessary. A cell was considered to have a single primary trunk if the initial thick apical trunk was at least 10 µm long before the first branchpoint.

**Statistical analysis**. No statistical methods were used to predetermine sample sizes, but our samples are similar to or exceed those reported in previous publications and that are generally employed in the field. The experimenter was aware of the experimental condition in the case of the electrophysiology-imaging experiments. All morphological analysis was performed blind to the electrophysiological phenotype of neurons. No explicit randomization method was used, but experiments comparing different cell types were typically interleaved. Statistical analysis was performed with Statistica software (Statsoft, Tulsa, OK, USA). Generally, nonparametric tests (Wilcoxon test for two paired groups, Mann–Whitney test for two unpaired groups, Friedman test for multiple paired groups, Kruskal–Wallis test for multiple unpaired groups, Spearman correlation) were used, which do not make assumptions about the distribution of data. In some analysis two-way ANOVA test was used with Tukey's test for post hoc comparisons; in these analyses the data passed the Levene test, except for the analysis on the number of trunks (Fig. 5i bottom). The specific test used for a given analysis is indicated in the text. All tests were two-tailed. For all dataset, $n$ represents number of cells, except in Fig. 4c where both the number of dendrites and number of cells is specified. Differences were considered significant when $p < 0.05$. In all figures except Fig. 1f, symbols and error bars represent mean ± s.e.m. *$p < 0.05$; **$p < 0.01$; ***$p < 0.001$.

**Reporting summary**. Further information on research design is available in the Nature Research Reporting Summary linked to this article.

## Data availability
The data and custom code that support the findings of this study are available from the corresponding author upon reasonable request. The source data underlying Figs. 1e–f, 2d–e, 3a, b, d, g, i, 4c, d, g, 5c–f, h–j, 6e, h, j, k and 7j, l, m are provided as a Source Data file.

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

## Acknowledgements

We thank B. Ujfalussy and B. Lükő for help in data analysis, and Z. Nusser, B. Ujfalussy, and all members of the Makara lab for useful discussions and comments. This work was supported by the Wellcome Trust (090915/Z/09/Z), the Lendület grant of the Hungarian Academy of Sciences (LP2011-012), the National Research, Development and Innovation Office of Hungary (ERC_HU_15 119023), the European Research Council (CoG 771849) and the International Research Scholar Program of the Howard Hughes Medical Institute (55008740).

## Author contributions

S.R.B., A.M., M.A., N.K. and J.K.M. performed electrophysiology and imaging experiments. A.L. performed and analyzed immunohistochemistry experiments. N.K. and Z.V.N. performed semiautomated morphological reconstruction. S.R.B., M.A., A.M. and J.K.M. analyzed electropysiology and imaging data. J.K.M. concieved and supervised the project and wrote the paper with input from all authors.

## Additional information

**Competing interests:** The authors declare no competing interests.

