## [Peer Review File · Nature Communications]

Reviewers' Comments:

Reviewer #1:

Remarks to the Author:

The authors argue that unlike in CA1 and L5 pyramidal neurons, where CSBs associated with global Ca events are triggered by co-activation of two distinct synaptic input pathways, a subset of CA3 neurons can generate CSBs with an associated global Ca event just by somatic depolarization or synaptic stimulation. They show that this is an intrinsic mechanism involving VGCCs and not a network-activity associated event. They explore the heterogeneity of this non-canonical CSB generation in CA3 population across the proximo-distal and radial axis, morphologies and gradients of HCN and Kv2.2 channels, finding the latter as principal regulators of CSB generation in CA3 cells.

Major

1. A central claim of the paper is that somatic depolarization or synaptic stimulation alone can trigger VGCC-mediated dendritic spikes that likely underly CSB firing in CA3 cells. This is a worthwhile and surprising result. While the authors do a good job of systematically characterizing it as an intrinsic phenomenon and looking at the influence of passive and active properties on its generation, it is disappointing that they do not investigate the dendritic generation of this phenomenon by whole cell recordings. Given the established prowess of the group in looking at active dendritic events in CA3 using dendritic whole cell recordings, one would have imagined that they would have taken a closer look at the dendritic electrogenesis of such a non-canonical VGCC event. It would be even more valuable if they can show how the dendritic depolarization changes with the peri-somatic Kv2.2 block using uncaging as the stimulus as depicted in fig 7f.

2. Previous studies (including some of the authors work) has shown that CA3 dendrites are not only more excitable but have a strong gradient/density of potassium channels like IA and inward rectifier. While the authors have a strong result describing the role of KV2.2, the role of potassium channels in dendritic electrogenesis would be more comprehensively addressed if they could block IA and KIR during CSB generation.

Minor:

1. Can the authors clarify that all the data in Fig1.D is using OGB-6f? It may not be perfect if the df/f values from 6f and 1 are pooled (which seems the case from methods) but that should be mentioned in the legend clearly.

2. Since the authors are measuring Calcium signals from distal tuft after somatic dye-loading, they should specify the diffusion time allowed for the dye to equilibrate.

3. The authors should mention the junction potential for their recording solutions.

4. The Immunolabeling data in Fig7 A is a bit difficult to visualize (especially in print). It would be helpful if the authors could overlay red and green channel line scan/intensity histogram across the figure to stress the colocalization.

Reviewer #2:

Remarks to the Author:

The manuscript investigates the physiological mechanisms that dictate Complex spike bursts in CA3

pyramidal cells. They identify intrinsic mechanisms (specifically regulation by HCN and Kv2.2 channels) that modulate CSB firing and, importantly, illuminate variability in such regulation within various regions of CA3. This brings to light important rules that may dictate how specific neurons are recruited to contribute to hippocampal circuits based on their respective dendritic/intrinsic properties. These rules may be applicable to various other regions of the brain.

In general, the authors provide a thorough explanation of CSBs and their modulation/generation along with their role in memory encoding in CA3 region of the hippocampus. The INTRODUCTION details synaptic mechanisms inducing bursting firing and the potential variation in various regions of CA3. They also provide useful discussion of CA3 microcircuits and pyramidal cell topography within these circuits which may confer variation in CSB generation among cell types. They make good use of published work here and provide necessary details while not straying too far away from the principle question as to cloud the big picture. The methods are generally carried out with care and they use appropriate techniques to address their questions. They are thorough in their screen of factors that may impact CSB firing, including location within the hippocampus, morphology, and passive and active membrane properties. Data from the approaches used are interpreted appropriately.

I think, overall they do a good job addressing questions that are of interest to the field. Identifying pyramidal cell heterogeneity in the hippocampus and how these heterogeneous cell types fit into network function is of general interest. This analysis provides useful information regarding interesting physiological variation in CA3 pyramidal cells that may prove insightful in deciphering hippocampal cellular and circuit mechanisms underlying associative memory. The discussion couples their findings with work from other labs to reveal the impact quite nicely. Some of their findings spark additional questions that would be interesting follow-ups, including the potential functional coupling of HCN channels (contradictory impact on excitability in distal CA3PCs) as well as additional intrinsic mechanisms contributing to CSB generation. Additionally, they postulate how experience may impact variation in CSB firing rate, so one could potentially address how the channels of interest and/or CSB rate in general are modulated by activity in various CA3PCs. Moreover, do these rules dictating CSBs occur in vivo? Thus, overall I think this is a that will be a nice contribution to the field.

I think there are a couple points that require attention, however. First, while use of nonparametric statistics for much of the analysis is acceptable, there are some experiments where I think additional replicates are needed to allow proper interpretation of the results. Two specific examples of this are the comparison of CSB rates (means) between CA3 and CA1 and the analysis of CSB rates in response to synaptic receptor blockade in CA3PCs. A large variation in sample size between neurons in these regions was utilized and it would be more appropriate to use parametric analysis for measurement of the means here. Moreover, the sample sizes are quite low for CSB rate measure in response to AMPA/NMDA and GABA_A block. In addition, there is some variation in observed responses in these conditions (particularly GluR block) and I think statistics need to be checked here. To strengthen the notion that network activity doesn't impact CSB rate, a higher sample size is needed to increase statistical power. I think providing a more detailed explanation of the workflow would help make clear the reason for use of the varied number of replicates.

Specific (mostly minor) concerns are detailed here:

Intro

1. INTRODUCTION heading missing
2. Line 134-should initially spell out 'stratum'

Results

3. Line 163: Explain the use of two different Ca²⁺ indicators? Perhaps some justification for this could be provided in the Methods section.

4. Line 166: depolarizing? I think more appropriate would be "depolarizing".
5. Figure 1B: define ddr
6. Figure 1E: Why is there consistent decrease in CSB rate by 3rd pulse followed by recovery by the 5th pulse with consistent interpulse interval in current injection trains? Touching on this mechanism in the Discussion would be useful with potential insights into specific mechanisms.
7. Line 180: high n in CA3 recordings vs low n in CA1. Especially if mean responses in CSB rate between CA3PCs and CA1PCs are being compared, I think increased sample size in CA1PCs and parametric analysis would increase statistical power.
8. Line 182: were current injections lower than 300pA used? The referenced figure only shows responses to 300pA and 600pA current injections and not any firing patterns in response to 200pA injections are indicated in the figure.
9. Figure 2: C,D 'yellow' color code appears orange.
10. Figure 3: Sample sizes for CSB rates in response to somatic current injection in the presence of TTX, Cd2+, GABARs, and GluRs are very low. Increasing these sample sizes would increase statistical power. Particularly for the GluR blockade, these data are difficult to interpret with such a low number of replicates as some variation in responses were observed.
11. Line 318: Does this stimulation reliably represent EPSCs evoked by local synaptic input? I think a better explanation of this stimulus paradigm and a rationale for its use in place of stimulation of proximal afferent input is necessary.
12. Figure 4F: I would suggest indicating 'summary of data recorded from RS cells' in the figure legend as opposed to simply 'summary of data'.
13. Line 421: 'Although' should be 'Although'
14. Figure 6C: in addition to listing 0-1 in relative positioning in CA3, I think it would be helpful to add proximal>distal along the x-axis to make it clear the Sag ratio along this axis (proximodistal axis).
15. It would be ideal to include additional techniques to manipulate channel activity in the hippocampus in addition to simply pharmacological ones, including RNAi; however, the channel blockers used here have been indicated to exhibit a high degree of specificity at the doses utilized in this study. Previous studies utilizing both HCN and Kv2.2 blockers in slice recordings indicate use of similar doses, and thus likely represent specific blockade of these channels. Thus the pharmacological experiments employed here likely have valid physiological connotation.

Discussion

16. Line 548: 'efficent' should be 'efficient'.

Methods

17. Line 727: Use of different Ca2+ chelators in internal solution and were pooled. What is the rationale for utilization of different Ca2+ indicators in this analysis?
18. I do not see any explanation regarding the duration of current injection in the dual-'synaptic' stimulation experiments measuring CSB rate in RC cells? Was the 600pA injection a constant injection paired with distal afferent stimulation or was the duration matched with a typical EPSC duration as to model physiological proximal synaptic input?
19. I think in general a little more detailed explanation of the workflow would be helpful. Presumably the neurons were characterized based on CSB firing rates in response to current injections, which were then followed by a host of different experiments on the patched neurons, ranging from glutamate uncaging to synaptic stimulation. These were chosen based on the initial neuronal firing phenotype in response to a train of current injections? This may be helpful to aid understanding of the differences in sample sizes among the various experiments as well. It would be assumed that a handful of neurons were chosen for each of the pharmacological experiments and these were independent of neurons that were used for two-photon uncaging and/or synaptic stimulation experiments.

Reviewer #3:

Remarks to the Author:

This is a very elegant study examining mechanisms of burst complex spike generation in CA3 pyramidal neurons. Given recent interest in these types of spikes and their putative role in plasticity and place field formation, the current study should be of wide interest. Moreover, relatively little is known about the mechanisms underlying these spikes in CA3. The authors have done a good job of documenting the heterogeneity in CA3 properties and have identified two interesting molecular candidates. Although it would have been of interest to know how differences in complex spike generation impact neuronal plasticity, the study as it stands represents an important contribution. I have only a few mostly technical concerns.

1. How confident are the authors that they record solely from CA3 cells rather than CA2 cells, especially in CA3a. The authors do not do a very thorough job of discussing this and need to indicate approximate boundaries between CA3a and CA2 in the figures. Did the authors always check to see that the neurons they recorded from had the thorny excrescences characteristic of CA3?

2. Figure 3a shows that NMDARs are not required to generate CSBs with somatic current injection. However, it is unclear whether these receptors help generate such spikes under more physiological conditions of synaptic stimulation. It would be interesting to see whether blockade of NMDARs alters CSBs using synaptic stimulation similar to figure 4.

3. The authors argue that the most effective way to elicit CSBs in their preparation is with combined synaptic stimulation in SLM and somatic depolarization (Figure 4). Again it would be of great interest to use stimulation of proximal MF inputs in place of current injection to examine such interactions under more physiological conditions.

4. As recent results show a great heterogeneity in strength of various synaptic inputs along the Proximodistal CA3 axis (Sun et al., 2017), this could provide another source of CSB heterogeneity.

5. The authors may wish to discuss the major implications of varying CSB propensity in CA3 PNs on plasticity and CA3 information processing, including such properties as pattern completion and sharp wave ripple generation.

Technical point 1. One potential worry in examining heterogeneity in slice preparations is whether some of the differences may reflect the severing of portions of the dendritic tree during slice preparation. The authors should provide more details about slice preparation, location of patched neurons (are they in the middle of the slice or near the surfaces) and should mention this as a possible complication.

Technical point 2: The parameters used to classify CSBs are a bit vague (see methods, page 35, l. 804), especially since the firing of CSBs compared to non-CSB spikes is the major point of the paper. It would be better to have clear cut-offs for AP frequency, amplitude reduction and ideally the OGB1 CA2+ signal should be a key parameter (or otherwise the authors should discuss why it is not used as such a criterion).

We would like to thank the reviewers for their positive opinion and thorough assessment of our work, as well as for suggestions for improvements. We have performed a number of new experiments and analyses to address their concerns, and added clarifications to the text as suggested. Below we provide a point-by-point response to the comments, and we hope that our responses will address all of the concerns.

Reviewer #1 (Remarks to the Author):

1. A central claim of the paper is that somatic depolarization or synaptic stimulation alone can trigger VGCC-mediated dendritic spikes that likely underly CSB firing in CA3 cells. This is a worthwhile and surprising result. While the authors do a good job of systematically characterizing it as an intrinsic phenomenon and looking at the influence of passive and active properties on its generation, it is disappointing that they do not investigate the dendritic generation of this phenomenon by whole cell recordings. Given the established prowess of the group in looking at active dendritic events in CA3 using dendritic whole cell recordings, one would have imagined that they would have taken a closer look at the dendritic electrogenesis of such a non-canonical VGCC event. It would be even more valuable if they can show how the dendritic depolarization changes with the peri-somatic Kv2.2 block using uncaging as the stimulus as depicted in fig 7f.

We carried out new whole-cell recordings from apical dendritic trunks of CA3PCs in proximal stratum radiatum (100-200 μm from the soma), allowing us to record closer to the assumed Ca^{2+} spike generation site(s) and still to be able to evoke CSBs by current injection. We used the same I_{inj} protocol as in our somatic experiments. In 8 out of 10 cells we found that I_{inj} evoked not only backpropagating APs (fast spikes with amplitude of 46.8 ± 4.3 mV and half width of 1.7 ± 0.1 ms, consistent with previous data from Kim et al., 2012 Nature Neurosci.), but also CSB-like voltage responses, which were driven by large, prolonged depolarizations following individual bAPs (ADP_{AP}). The maximal ADP_{AP} amplitude during the CSBs was approximately twofold larger when measured with dendritic recordings than that measured in independent somatic recordings. The larger ADP_{AP} amplitude in dendrite versus soma was in contrast with the smaller dendritic amplitude of the first action potential (which propagates from the soma). Altogether these results suggest that during CSBs the main charge source for the slow depolarization is produced in the dendrites. In addition, when we bath applied TTX to block bAPs, isolated Ca^{2+} spikes could be readily evoked by moderate I_{inj} into the dendrite (tested in 5 cells). These new data are included in the revised manuscript as Figure 3H-I.

Due to the complex apical dendritic arbour structure of CA3PCs (which have multiple thick trunks with potentially different excitabilities), precise determination of the location of Ca^{2+} spike generation and its spread, as well as the contribution of an individual trunk or dendritic area to the CSB is a challenging but exciting task that requires technically difficult multisite recordings. Furthermore, these properties are expected to depend also on topographic location in CA3. We are working actively towards addressing these issues in a separate, comprehensive study.

2. Previous studies (including some of the authors work) has shown that CA3 dendrites are not only more excitable but have a strong gradient/density of potassium channels like IA and inward rectifier. While the authors have a strong result describing the role of KV2.2, the role of potassium channels in dendritic electrogenesis would be more comprehensively addressed if they could block IA and KIR during CSB generation.

In this study we have specifically focused on the roles of HCN and Kv2.2 channels not only because they are present in dendrites, but because they are expressed in a prominent proximodistal gradient along CA3, possibly explaining some of the regional differences observed in CSB propensity. However, as CSBs are produced by a complex interplay of dendritic and somatic voltage-dependent conductances, we fully agree with the Reviewer in that other conductances, including other types of K^+ currents, may influence the generation of CSBs. In fact, we assume that inhibition of any channel

(including K_{IR} and I_A channels) that regulates somatic AP properties and/or the electrotonic spread of depolarization will have some impact on CSB propensity.

Our previous work revealed a prominent role for K_{IR} channels in regulating subthreshold NMDA spike decay and temporal summation, with less impact of I_A current (Makara & Magee 2013 Neuron). Per the Reviewer's suggestion, we performed new experiments in which we aimed to test the effect of K_{IR} on CSB propensity by bath applying low concentration of Ba^{2+} ($30 \mu M$). Ba^{2+} increased the input resistance as expected (Makara & Magee 2013 Neuron); however, the effect of K_{IR} blockade on CSB rate was difficult to interpret, as illustrated in Figure R1 below.

In PCs of the CA3c region, Ba^{2+} clearly increased CSB propensity ($n=9$ cells, Figure R1A and C). However, in PCs in CA3a-b, the firing behaviour in response to the I_{inj} protocol changed drastically, as most cells (4 out of 6) did not repolarize properly between the sequential pulses and rather fired persistently over several hundred milliseconds (see example cell in Figure R1B). In these pulses the occurrence of CSBs could not be assessed reliably. Without proper quantitative analysis of CSB rate possible, we feel that this experiment would not add significantly to the manuscript.

Figure R1. Effect of $30 \mu M Ba^{2+}$ on CSB rate. A-B) Representative somatic voltage (black) and dendritic Ca^{2+} traces (green) recorded from a CA3c PC (A) and a CA3a PC (B) using 300 pA I_{inj} protocol under control conditions (upper traces) and after >10 min bath application of $30 \mu M Ba^{2+}$ (lower traces). Note the persistent depolarization response in Ba^{2+} in (B), rendering at least two pulses uninterpretable for CSB rate measurement. C) Summary of the effect of $30 \mu M Ba^{2+}$ on CSB rate by 300 and 600 pA I_{inj} in CA3c PCs ($n=9$).

Investigation of the effect of I_A channels on CSB rate is also problematic with pharmacological tools. In previous work we have inhibited these channels with $200 \mu M Ba^{2+}$ to study their role in synaptic integration (Losonczy et al., 2008 Nature; Makara et al. 2009 Nat. Neurosci; Makara & Magee 2013 Neuron); however the above problem occurring in $30 \mu M Ba^{2+}$ would certainly be even more exacerbated with higher concentrations. With the other widely used I_A inhibitor 4-AP, we encountered spontaneous epileptiform population discharges in previous work (Makara & Magee 2013 Neuron), consistent with other reports (e.g. Voskuyl & Albus 1985; Perreault & Avoli 1992). Therefore, in the past we coapplied 4-AP with TTX (Makara & Magee 2013 Neuron), but this treatment cannot be employed when studying CSBs. For these reasons, we refrained from evaluating the role of I_A in regulating CSB rate in this study.

As we agree that the potential impact of other conductances should be clearly pointed out in the paper, we elaborated on the possible roles of other K^+ channels in regulating CSB propensity in the Discussion (p.21).

Minor:

1. Can the authors clarify that all the data in Fig1.D is using OGB-6f? It may not be perfect if the df/f values from 6f and 1 are pooled (which seems the case from methods) but that should be mentioned in the legend clearly.

We regret that this was not clear in the first version of our manuscript. The data in Figure 1D are the $\Delta F/F$ and ADP measurements from the representative experiment shown in Fig. 1A-C (50 pulses in 10 traces), therefore all data were measured with OGB-6F. We have explicitly state this now in the legend.

2. Since the authors are measuring Calcium signals from distal tuft after somatic dye-loading, they should specify the diffusion time allowed for the dye to equilibrate.

We waited >20 minutes after establishing the whole-cell configuration to allow sufficient diffusion of the fluorescent dyes to the distal dendrites. We included this information into the Methods (p.27).

3. The authors should mention the junction potential for their recording solutions.

The liquid junction potential (+10 mV) was not corrected for; this has been clarified now in the Methods (p.26).

4. The Immunolabeling data in Fig7 A is a bit difficult to visualize (especially in print). It would be helpful if the authors could overlay red and green channel line scan/intensity histogram across the figure to stress the colocalization.

We have increased the contrast of our images, and included line scan profiles in the revised Figure 7, as suggested by the Reviewer.

Reviewer #2 (Remarks to the Author):

The manuscript investigates the physiological mechanisms that dictate Complex spike bursts in CA3 pyramidal cells. They identify intrinsic mechanisms (specifically regulation by HCN and Kv2.2 channels) that modulate CSB firing and, importantly, illuminate variability in such regulation within various regions of CA3. This brings to light important rules that may dictate how specific neurons are recruited to contribute to hippocampal circuits based on their respective dendritic/intrinsic properties. These rules may be applicable to various other regions of the brain.

In general, the authors provide a thorough explanation of CSBs and their modulation/generation along with their role in memory encoding in CA3 region of the hippocampus. The INTRODUCTION details synaptic mechanisms inducing bursting firing and the potential variation in various regions of CA3. They also provide useful discussion of CA3 microcircuits and pyramidal cell topography within these circuits which may confer variation in CSB generation among cell types. They make good use of published work here and provide necessary details while not straying too far away from the principle question as to cloud the big picture. The methods are generally carried out with care and they use appropriate techniques to address their questions. They are thorough in their screen of factors that may impact CSB firing, including location within the hippocampus, morphology, and passive and active membrane properties. Data from the approaches used are interpreted appropriately.

I think, overall they do a good job addressing questions that are of interest to the field. Identifying pyramidal cell heterogeneity in the hippocampus and how these heterogeneous cell types fit into

network function is of general interest. This analysis provides useful information regarding interesting physiological variation in CA3 pyramidal cells that may prove insightful in deciphering hippocampal cellular and circuit mechanisms underlying associative memory. The discussion couples their findings with work from other labs to reveal the impact quite nicely. Some of their findings spark additional questions that would be interesting follow-ups, including the potential functional coupling of HCN channels (contradictory impact on excitability in distal CA3PCs) as well as additional intrinsic mechanisms contributing to CSB generation. Additionally, they postulate how experience may impact variation in CSB firing rate, so one could potentially address how the channels of interest and/or CSB rate in general are modulated by activity in various CA3PCs. Moreover, do these rules dictating CSBs occur *in vivo*? Thus, overall I think this is a that will be a nice contribution to the field.

I think there are a couple points that require attention, however. First, while use of nonparametric statistics for much of the analysis is acceptable, there are some experiments where I think additional replicates are needed to allow proper interpretation of the results. Two specific examples of this are the comparison of CSB rates (means) between CA3 and CA1 and the analysis of CSB rates in response to synaptic receptor blockade in CA3PCs. A large variation in sample size between neurons in these regions was utilized and it would be more appropriate to use parametric analysis for measurement of the means here. Moreover, the sample sizes are quite low for CSB rate measure in response to AMPA/NMDA and GABA_A block. In addition, there is some variation in observed responses in these conditions (particularly GluR block) and I think statistics need to be checked here. To strengthen the notion that network activity doesn't impact CSB rate, a higher sample size is needed to increase statistical power. I think providing a more detailed explanation of the workflow would help make clear the reason for use of the varied number of replicates.

As we detail below, we performed a number of additional recordings to increase the sample sizes in all experiments pointed out by the Reviewer, and the results strengthened our previous conclusions. Wherever possible, we performed nonparametric rather than parametric analysis on CSB rates, because the distribution of these data was not normal, and even with various transformations we could not get closer to satisfy the conditions for parametric statistical tests. We nevertheless performed the statistical comparisons also using parametric tests, which yielded very similar results.

Specific (mostly minor) concerns are detailed here:

Intro

1. *INTRO heading missing*

Corrected.

2. *Line 134-should initially spell out 'stratum'*

Corrected.

Results

3. *Line 163: Explain the use of two different Ca²⁺ indicators? Perhaps some justification for this could be provided in the Methods section.*

The amplitude of CSB-associated dendritic Ca²⁺ signals varied substantially across individual CA3PCs, making it difficult to choose one Ca²⁺ indicator optimal to use in all cells. We initially applied the high affinity dye OGB-1 because we did not want to miss Ca²⁺ events in the distal dendrites. Using OGB-1, CSBs were typically accompanied by Ca²⁺ signals that were much larger than occasional small Ca²⁺ signals evoked by backpropagating action potentials. We had two reasons to perform experiments also with a low affinity indicator (OGB-6F): first, in some CA3PCs the large CSB-associated dendritic

Ca²⁺ signals caused saturation of OGB-1; second, we wished to examine the possibility that Ca²⁺ buffering by the high affinity OGB-1 may influence the CSB rate *per se* for example by limiting the recruitment of Ca²⁺-activated K⁺ channels. This latter concern was excluded, because we found similar CSB rates with both dyes (100 μM, Mann-Whitney test for 300 pA: p=0.190, for 600 pA: 0.812), and recorded CSB_H cells also with the non- Ca²⁺ sensitive dye Alexa488 (as mentioned in the Methods; n=5). While saturation was not a problem with OGB-6F, the signal-to-noise ratio was less favourable in CA3PCs with smaller CSB-associated Ca²⁺ signals. Altogether, since we only utilized the occurrence, but not an absolute amplitude value of large dendritic Ca²⁺ signals as an aid to identify CSBs, and because we found no dye-dependent difference in CSB rate, we considered it justified to include experiments in the study with both indicators. We also included the explanation in the Methods (p. 26).

4. Line 166: *depolarizing?* I think more appropriate would be “depolarizing”.

Corrected.

5. Figure 1B: *define ddr*

Corrected.

6. Figure 1E: *Why is there consistent decrease in CSB rate by 3rd pulse followed by recovery by the 5th pulse with consistent interpulse interval in current injection trains? Touching on this mechanism in the Discussion would be useful with potential insights into specific mechanisms.*

Indeed, as correctly pointed out by the reviewer, in CSB_H cells (Fig. 1E) we saw a consistent temporal pattern of CSB rate during the series of I_{inj} pulses, with the lowest CSB rate on the 3rd pulse. We have currently no explanation for this pattern, and can only speculate that it might be the result of a complex interplay of different ion conductances involved in CSB generation, being activated/inactivated on a time scale (seconds) corresponding to the duration of the whole I_{inj} pulse series rather than that of a single 100-ms-long pulse. Because we have no solid explanation for this phenomenon, we would prefer to stick to a simple presentation of the data without speculations about the mechanisms.

7. Line 180: *high n in CA3 recordings vs low n in CA1. Especially if mean responses in CSB rate between CA3PCs and CA1PCs are being compared, I think increased sample size in CA1PCs and parametric analysis would increase statistical power.*

We have performed additional experiments in CA1PCs, increasing the number of cells to 26. Now comparing the CA3 and CA1 PC groups with the nonparametric Mann-Whitney test yielded highly significant p values both for 600 pA I_{inj} (p=0.00015) as well as for 300 pA I_{inj} (p=0.008). Performing the parametric Student's t-tests also yielded significant differences (600 pA: p=0.0005, 300 pA: p=0.003); however, we refrained from performing parametric analysis because the distribution of the CSB rate data was not normal (see Figure 2C, Shapiro-Wilks test p<0.0001 both at 300 pA and 600 pA), the Levene test indicated strong violation of the requirement for homogeneity of variances (p<0.0001), and even with various transformations we could not get closer to satisfy the conditions for parametric statistical tests. We updated Figure 1F in the revised manuscript.

8. Line 182: *were current injections lower than 300pA used? The referenced figure only shows responses to 300pA and 600pA current injections and not any firing patterns in response to 200pA injections are indicated in the figure.*

The Reviewer is correct that the referenced figure only shows responses to 300 pA I_{inj}; therefore we amended the text to avoid confusion. Although we did not apply 200 pA I_{inj} pulses systematically in all cells, in a fraction of CSB_H cells (n=15) we also measured the CSB rate with 200 pA and found that

approximately half of these neurons (n=7) displayed CSBs even at 200 pA. This information has now been included in the Methods (p.27).

9. Figure 2: C,D 'yellow' color code appears orange.

We corrected the colour description to "orange".

10. Figure 3: Sample sizes for CSB rates in response to somatic current injection in the presence of TTX, Cd²⁺, GABARs, and GluRs are very low. Increasing these sample sizes would increase statistical power. Particularly for the GluR blockade, these data are difficult to interpret with such a low number of replicates as some variation in responses were observed.

We carried out additional experiments to increase the sample size of these datasets: we have now n=10 for GluR blockers (revised Figure 3A), n=11 for GABAR blockers (revised Figure 3B), n=11 for TTX (revised Figure 3C) and n=7 for TTX+Cd²⁺ (revised Figure 3D) experiments. The additional data strengthened our previous conclusions; in particular, we did not observe any new cases where the CSB rate changed substantially upon application of GluR blockers.

11. Line 318: Does this stimulation reliably represent EPSCs evoked by local synaptic input? I think a better explanation of this stimulus paradigm and a rationale for its use in place of stimulation of proximal afferent input is necessary.

We agree that a simple square-shaped depolarizing I_{inj} cannot faithfully represent afferent activity patterns, but we argue that sustained depolarization with similar overall duration may occur under naturalistic conditions. Unfortunately, *in vivo* intracellular recordings of the postsynaptic effect of natural MF activity in CA3PCs in awake rodents are not yet available. Our rationale that a 100-ms-long I_{inj} stimulus could coarsely mimic perisomatic depolarization evoked by *in vivo* mossy fibre activity is based on extrapolations from *in vivo* firing activity and *in vitro* measured intracellular postsynaptic responses. Dentate gyrus granule cells can fire bursts of 3-5 APs at ~40-100 Hz (Henze et al., 2002 Nat. Neurosci., Pernia-Andrade et al., 2014 Neuron). Because unitary EPSPs evoked by MFs in CA3PCs decay relatively slowly ($\tau_{decay} \sim 130$ ms, Vyleta et al., 2016 eLife), even a moderate burst of 3 APs at 50 Hz is expected to cause depolarization of the CA3PC for >100 ms without returning to baseline V_m (Vyleta et al., 2016 eLife; Chamberland et al., 2018 PNAS). Furthermore, EPSPs evoked in the thin proximal dendrites by recurrent collaterals of other CA3PCs activated by the MF are expected to further contribute to the depolarization. Therefore, the combined activation of MFs and the downstream CA3 recurrent circuitry could lead to a relatively sustained (albeit temporally varying) depolarization in the perisomatic dendritic compartment. In addition to these considerations, a previous landmark study on the role of CSBs in CA1PCs also used square-pulse somatic I_{inj} to mimic perisomatic depolarization, and combined it with distal synaptic stimulation to evoke dendritic plateaus *in vitro* (Bittner et al., 2015 Nature Neurosci.). We added reference to this latter work.

12. Figure 4F: I would suggest indicating 'summary of data recorded from RS cells' in the figure legend as opposed to simply 'summary of data'.

Corrected as suggested.

13. Line 421: 'Although' should be 'Although'

Corrected.

14. Figure 6C: in addition to listing 0-1 in relative positioning in CA3, I think it would be helpful to add proximal>distal along the x-axis to make it clear the Sag ratio along this axis (proximodistal axis).

We added 'proximal to distal' to the axis title as suggested.

15. *It would be ideal to include additional techniques to manipulate channel activity in the hippocampus in addition to simply pharmacological ones, including RNAi; however, the channel blockers used here have been indicated to exhibit a high degree of specificity at the doses utilized in this study. Previous studies utilizing both HCN and Kv2.2 blockers in slice recordings indicate use of similar doses, and thus likely represent specific blockade of these channels. Thus the pharmacological experiments employed here likely have valid physiological connotation.*

We thank for the comment and agree with the Reviewer that additional approaches using genetic manipulations will be useful to corroborate our results in future research (as well as allowing to study the roles of HCN and Kv2.2 in CA3PCs *in vivo*). Nevertheless, both ZD7288 and guangxitoxin have been used by multiple investigators and are generally accepted to be specific inhibitors of HCN and Kv2 channels, respectively.

Discussion

16. *Line 548: 'efficent' should be 'efficient'.*

Corrected.

Methods

17. *Line 727: Use of different Ca²⁺ chelators in internal solution and were pooled. What is the rationale for utilization of different Ca²⁺ indicators in this analysis?*

As stated above, one concern that led us to use fluorescent dyes with different Ca²⁺ affinities was the possibility that Ca²⁺ binding and buffering by a high affinity dye may influence the CSB rate, for example by limiting the recruitment of Ca²⁺-activated K⁺ channels. This concern led us also to measure CSB rate in a group of cells using the non-Ca²⁺ sensitive dye Alexa488 (in these experiments CSBs were identified solely based on the somatic electrophysiological signature). Because we found similar CSB rates with both low and high affinity dyes (see above), and because we recorded CSB_H cells even with Alexa488, we excluded the possibility that Ca²⁺ buffering significantly affected CSB rate, and therefore the results were pooled. This rationale has been included in the revised Methods (p.26).

As also mentioned above, we found that the amplitude of CSB-associated dendritic Ca²⁺ signals was highly variable across different individual CA3PCs, which made it difficult to choose one Ca²⁺ indicator that would be optimal to use in all cells. The high affinity dye OGB-1 (applied initially) confirmed that CSBs were accompanied by Ca²⁺ signals in distal dendrites; however, in some CA3PCs the large signals saturated the dye. On the other hand, while saturation did not occur with the lower affinity OGB-6F, the signal-to-noise ratio was less favourable in cells with smaller CSB-associated Ca²⁺ signals. Altogether, since we only utilized the occurrence of dendritic Ca²⁺ signals, but not their amplitude values as an aid to identify CSBs, we felt it justified to include experiments in the study with both dyes.

18. *I do not see any explanation regarding the duration of current injection in the dual-'synaptic' stimulation experiments measuring CSB rate in RC cells? Was the 600pA injection a constant injection paired with distal afferent stimulation or was the duration matched with a typical EPSC duration as to model physiological proximal synaptic input?*

The temporal pattern of the 600 pA I_{inj} in the pairing experiments was exactly the same as the simple I_{inj} protocol: i.e., each of the five 100-ms-long pulses was paired with a burst of distal synaptic stimulation. The protocol is shown in Figure 4F.

19. *I think in general a little more detailed explanation of the workflow would be helpful. Presumably*

the neurons were characterized based on CSB firing rates in response to current injections, which were then followed by a host of different experiments on the patched neurons, ranging from glutamate uncaging to synaptic stimulation. These were chosen based on the initial neuronal firing phenotype in response to a train of current injections? This may be helpful to aid understanding of the differences in sample sizes among the various experiments as well. It would be assumed that a handful of neurons were chosen for each of the pharmacological experiments and these were independent of neurons that were used for two-photon uncaging and/or synaptic stimulation experiments.

We apologize that the workflow was not clear in the original manuscript. The experiments were indeed conducted as the reviewer described. We included a section in the Methods to provide readers with the information about the workflow (p.27).

Reviewer #3 (Remarks to the Author):

This is a very elegant study examining mechanisms of burst complex spike generation in CA3 pyramidal neurons. Given recent interest in these types of spikes and their putative role in plasticity and place field formation, the current study should be of wide interest. Moreover, relatively little is known about the mechanisms underlying these spikes in CA3. The authors have done a good job of documenting the heterogeneity in CA3 properties and have identified two interesting molecular candidates. Although it would have been of interest to know how differences in complex spike generation impact neuronal plasticity, the study as it stands represents an important contribution. I have only a few mostly technical concerns.

1. How confident are the authors that they record solely from CA3 cells rather than CA2 cells, especially in CA3a. The authors do not do a very thorough job of discussing this and need to indicate approximate boundaries between CA3a and CA2 in the figures. Did the authors always check to see that the neurons they recorded from had the thorny excrescences characteristic of CA3?

We carefully inspected the 2P image of the proximal dendrites in every recorded neuron at the beginning of the experiment in order to confirm the presence of thorny excrescences (as well as numerous small spines on thinner dendrites). Occasional athorny cells were encountered only at the CA2 border; these recordings were terminated. Therefore we are confident that all cells recorded in this study were CA3 pyramidal cells. The border between CA2 and CA3a was considered to be at 200 μm from the border between CA1 and CA2 (which was identified as the area where somata became larger and elongated, and str. pyramidale widened, as shown on Figure 5A). We included clarification in the revised Methods (p.27, 32).

2. Figure 3a shows that NMDARs are not required to generate CSBs with somatic current injection. However, it is unclear whether these receptors help generate such spikes under more physiological conditions of synaptic stimulation. It would be interesting to see whether blockade of NMDARs alters CSBs using synaptic stimulation similar to figure 4.

We thank the Reviewer for the suggestion, this is indeed an interesting point. We performed new experiments to examine the contribution of NMDARs to CSB generation by local synaptic stimulation. In particular, we tested CSB rate in n=8 CSB cells using glutamate uncaging at the same synapse pattern under control conditions and after bath application of D-AP5. We found that 1) D-AP5 decreased EPSP summation (as expected), and 2) it moderately reduced the CSB rate by suprathreshold synaptic stimulation, mostly in cells that had CSB rate of <1. This indicates that NMDARs can indeed facilitate CSB generation in CSB cells, by increasing the amplitude and prolonging the time course of the EPSP. The effect of AP5 is presented in a new panel (Figure 4D) and in the text (p.11). We also updated Figure

4C by including the additional data on uncaging-evoked CSB rate measured under control conditions in these experiments.

3. The authors argue that the most effective way to elicit CSBs in their preparation is with combined synaptic stimulation in SLM and somatic depolarization (Figure 4). Again it would be of great interest to use stimulation of proximal MF inputs in place of current injection to examine such interactions under more physiological conditions.

As Figure 4 shows, the requirements for evoking CSB were different in RS and CSB cells: combined somatic depolarization and distal synaptic activity was more efficient *in RS cells*, whereas *in CSB_H cells* local correlated synaptic activity was sufficient.

We expect that in RS cells, the combination of distal synaptic activity with any source of perisomatic depolarization would be efficient to induce CSBs. While MFs could indeed represent a candidate proximal pathway, it is not the only one - recurrent collaterals also form proximal synapses on perisomatic basal and apical dendrites, and could provide another, just as important depolarization source. The dominant perisomatic input may be heterogeneous in different CA3PCs: e.g. the most strongly bursting cells receive little if any MF innervation (Hunt et al. 2018 Nature Neurosci.) and the contribution of MF and CA3 inputs may be different in different CA3 subregions (Sun et al., 2017 Neuron). In addition, isolated activation of MFs is challenging even if possible with extracellular stimulation in acute slices (Henze et al., 2000, Neuroscience). Due to these considerations we did not attempt to dissect which of the possible combinations of proximal and distal synaptic inputs would most efficiently trigger CSBs.

4. As recent results show a great heterogeneity in strength of various synaptic inputs along the Proximodistal CA3 axis (Sun et al., 2017), this could provide another source of CSB heterogeneity.

Indeed, we agree with the Reviewer that in addition to the heterogeneity in intrinsic properties, gradients of synaptic strength and connectivity (together with potential differences in the activity of the afferent neurons) can contribute to the heterogeneity of CSB propensity. We have included reference to this point in the Discussion (p.20).

5. The authors may wish to discuss the major implications of varying CSB propensity in CA3 PNs on plasticity and CA3 information processing, including such properties as pattern completion and sharp wave ripple generation.

We have discussed these questions to some extent in the last three paragraphs of the original manuscript, in particular the possible relevance of the proximodistal heterogeneity of CSB propensity for pattern separation/completion, and we speculated on the link of CSBs with synaptic plasticity. We added further thoughts on the potential role of CSB_H cells in sharp wave ripple generation and the possibility of heterogeneous CA3PCs forming parallel information processing networks (p.23-24).

Technical point 1. One potential worry in examining heterogeneity in slice preparations is whether some of the differences may reflect the severing of portions of the dendritic tree during slice preparation. The authors should provide more details about slice preparation, location of patched neurons (are they in the middle of the slice or near the surfaces) and should mention this as a possible complication.

We agree that this is a potential concern. Indeed, for this reason we took great care to perform our experiments on neurons with as complete dendritic arbour as possible. This is especially important in the case of CA3PCs which have a widely branching apical arbour. To access neurons with mostly preserved dendritic tree, in every slice we selected a region within CA3 where the apical trunks of most neurons were oriented slightly downwards (directed into the tissue), and we patched somata located at least 50 µm deep. We inspected every neuron after loading with fluorescent dyes, and excluded

cells in which any of the main apical trunks in str. radiatum was cut. We included these details in the revised Methods (p.25-27).

Technical point 2: The parameters used to classify CSBs are a bit vague (see methods, page 35, l. 804), especially since the firing of CSBs compared to non-CSB spikes is the major point of the paper. It would be better to have clear cut-offs for AP frequency, amplitude reduction and ideally the OGB1 CA2+ signal should be a key parameter (or otherwise the authors should discuss why it is not used as such a criterion).

In most cells the occurrence of CSBs by somatic current injection was easily identified by visual inspection of voltage and Ca^{2+} traces. In a set of neurons ($n=24$) we also confirmed the reliability of the visual categorization with numerical parameters. We could not use the same uniform set of cut-off values to identify CSBs in all cells, because not only the CSB rate but also the properties of CSBs (i.e. duration, amplitude and kinetics of the ADP, number of APs involved) were highly heterogeneous across different CA3PCs, along with heterogeneity in the rate and adaptation pattern of simple APs. The kinetics of CSBs could also change with the number of I_{inj} pulses in the series. Nevertheless, the firing pattern was characteristic in a given cell, and thus it was possible to rely on a combination of parameters to separate pulses with and without CSBs in each individual cell, as explained below. Tested on 600 depolarization pulses from 24 cells, the agreement of visual and numerical categorization was ~99%, in our opinion satisfactory to allow relying on visual categorization.

The following parameters were taken into account for the numerical analysis:

1) Electrophysiological parameters:

- Amplitude of the slow ADP evoking sequential AP(s),
- Maximal decrease of AP peak amplitude, and
- Maximal relative increase of AP frequency.

The product of these three parameters, which we term “CSB score”, well separated pulses into two major groups (with and without CSBs) within a given cell. We provide examples in Figure R2 below.

2) Ca^{2+} signal:

Time-locked distal dendritic Ca^{2+} signals (significantly larger than Ca^{2+} signals evoked by pulses with only simple backpropagating APs) were typically associated with CSBs, as shown in the manuscript figures as well as in Figure R2A. Nevertheless, Ca^{2+} signals were used as a complementary rather than mandatory criterion to identify CSBs for the following reasons:

- Dynamic range issues due to cell-to-cell heterogeneity of Ca^{2+} signal amplitudes:
 - a) Using OGB-1, in some cells large CSB-associated Ca^{2+} signals saturated the dye, masking the Ca^{2+} signal of the (electrophysiologically evident) CSB on the following I_{inj} pulse within the series (see example in Figure R2B).
 - b) Using OGB-6F, in cells with smaller CSB-associated Ca^{2+} signals the signal-to-noise ratio was relatively poor. In these cases, if the electrophysiological signature of CSBs at the soma was unambiguous, we identified CSBs without taking the Ca^{2+} signal into account (see example in Figure R2C).
- Other considerations:
 - a) In many cells the first I_{inj} pulse evoked more APs and longer CSBs than later pulses, with larger corresponding dendritic Ca^{2+} responses (both for CSBs and simple APs), making it difficult to define a uniform threshold Ca^{2+} signal amplitude for CSBs.
 - b) Some experiments were conducted with a Ca^{2+} insensitive dye (Alexa488).

Ca^{2+} signals were therefore usually used to confirm our categorization of pulses with or without CSBs, and the presence or absence of the dendritic Ca^{2+} signal was decisive in cases where the electrophysiological identification of the CSB remained questionable.

Figure R2. Examples for CSB identification based on electrophysiological parameters (CSB score) and dendritic Ca^{2+} signals in different CA3PCs.

Left panels: z-stacks indicating linescan position. Middle panels: two example sets of voltage and simultaneous dendritic Ca^{2+} recordings in response to 600 pA I_{inj} pulses. Red asterisks label CSBs that are not directly preceded by pulse with CSB; blue asterisks label CSBs where the preceding pulse evoked CSB. Right panels: CSB score and Ca^{2+} signal measured in 5 traces (25 pulses total). Black circles indicate pulses without CSB; coloured circles indicate pulses with CSBs, as explained above. CSB score is presented on a logarithmic scale; dashed grey lines emphasize CSB score of 100 and 400 on all panels.

A) A typical simple case using OGB-1. B) Ca^{2+} signals by OGB-1 can saturate on consecutive pulses with CSBs. Note the small Ca^{2+} signal amplitudes despite high CSB scores belonging to the second pulse evoking CSB (blue). C) A recording where OGB-6F Ca^{2+} signals are small, despite electrophysiologically unequivocal CSBs. The CSB score still separates pulses with and without CSBs. D) In some cells, CSBs are relatively short and have small CSB scores, but are clearly associated with dendritic Ca^{2+} signals. Pulses with and without CSBs are still well separated using the CSB score and the Ca^{2+} signal.

Note that a single set of parameters cannot be used in all four neurons to define all CSBs.

In the uncaging experiments, detection of CSBs was based on the electrophysiological profile alone, because synaptic stimulation produced large local Ca^{2+} responses *per se*. In the revised manuscript we reanalysed our recordings with more strict criteria for CSBs, i.e. only responses with a high-frequency burst of at least 3 APs with progressively reduced peak amplitudes riding on a slow ADP were considered as CSBs. Because the kinetics of the voltage response produced by the gamma burst uncaging pattern was substantially different from that evoked by I_{inj} (gradual depolarization evoking fewer APs), the above numerical analysis was not applied here.

We have included a detailed explanation of CSB identification in the Methods section (p.30-31).

Reviewers' Comments:

Reviewer #1:

Remarks to the Author:

The data provided by authors in Fig.R1 looks interesting given the mechanistic focus of the authors on CSB propensity especially in CA3c region (Compare R1C to fig.7F, one of the main findings of this study). I disagree with the authors that the finding would not significantly add to the manuscript and would encourage them to at least mention the effect of low Ba²⁺ in their results even if they decide not to include the dataset based on the lack of quantification of CSB rate for CA3a-b.

The authors have adequately addressed all my other concerns and I commend this manuscript for publication.

Reviewer #2:

Remarks to the Author:

The authors went above and beyond in addressing the original concerns. The addition of dendritic recordings in order to further address the role of dendritic events, alone, in triggering CSB firing augments the rest of the data and enhances the impact of the findings. They also added experiments to screen additional intrinsic factors that may play a role in the generation of CSB firing.

Each concern was addressed appropriately.

Therefore, I think that the additions to the manuscript and rebuttals from the authors are satisfactory.

Reviewer #3:

Remarks to the Author:

The authors have done a commendably complete job of addressing my initial concerns and critiques. I have no further concerns to address.

We would like to thank all of the Reviewers for their thorough evaluation of our work and for supporting publication of our manuscript.

Point-by-point response

Reviewer #1

The data provided by authors in Fig.R1 looks interesting given the mechanistic focus of the authors on CSB propensity especially in CA3c region (Compare R1C to fig.7F, one of the main findings of this study). I disagree with the authors that the finding would not significantly add to the manuscript and would encourage them to at least mention the effect of low Ba²⁺ in their results even if they decide not to include the dataset based on the lack of quantification of CSB rate for CA3a-b.

The authors have adequately addressed all my other concerns and I commend this manuscript for publication.

We thank the Reviewer for the constructive feedback and appreciation of our revisions.

Inferring from results with data not presented is not allowed by Nature Communications policy, therefore only the results in CA3cPCs might be included. However, although proper quantification was problematic, Ba²⁺ clearly had an effect on PCs in CA3a-b to increase CSB propensity (as Figure R1B showed). The fact that several CA3aPCs remained depolarized between I_{inj} pulses in Ba²⁺ may in fact suggest that IRK channels might play a more dominant role in distal CA3. Our concern is that showing only CA3c results but not the full picture could lead to a false impression that inward rectifying K⁺ channels specifically regulate CSBs in CA3c. In addition, since IRK channels fundamentally control basic electrical properties of neurons, it would be surprising if their blockade had no effect on CSB propensity, and thus, in our opinion the results with Ba²⁺ are rather confirmative.

For the above reasons we would prefer to remain at simply mentioning the likely role of IRK channels in regulation of CSB propensity in the Discussion.